# Leveraging Randomness in Model and Data Partitioning for Privacy Amplification

**Andy Dong** [1]    **Wei-Ning Chen** [2]    **Ayfer Ozgur** [1]

## Abstract

We study how inherent randomness in the training process—where each sample (or client in federated learning) contributes to only a randomly selected portion of training—can be leveraged for privacy amplification. This includes (1) model partitioning, where a sample updates only a subset of the model parameters, and (2) data partitioning, where a sample participates in only a subset of training iterations. We apply our framework to model parallelism in federated learning, where each client updates a randomly selected subnetwork to reduce memory and computational overhead, and show that existing methods, e.g. model splitting or dropout, provide a significant privacy amplification gain not captured by previous privacy analysis techniques. Additionally, we introduce Balanced Iteration Subsampling, a new data partitioning method where each sample (or client) participates in a fixed number of training iterations. We show that this method yields similar or stronger privacy amplification than Poisson (i.i.d.) sampling of data (or clients). Our results demonstrate that randomness in the training process, which is structured rather than i.i.d. and interacts with data in complex ways, can be systematically leveraged for significant privacy amplification.

## 1. Introduction

Ensuring user privacy in machine learning is critical, especially in applications involving sensitive data such as healthcare, finance, and social networks. Differential privacy (DP) (Dwork et al., 2006; Dwork, 2006) provides a rigorous mathematical framework to quantify and limit data leakage, offering strong protections even against adversaries with auxiliary information. The standard way to achieve DP is to inject randomness into the training process, typically by adding noise—such as in DP-SGD (Abadi et al., 2016)—which provides formal privacy guarantees. However, this comes at a cost: injecting more noise strengthens privacy protections but often degrades model performance.

In this paper, we ask: instead of relying solely on large noise injections to achieve strong privacy guarantees, can we leverage the inherent randomness already present in the training process? Many widely used techniques, such as dropout (which helps mitigate overfitting) and model parallelism (which reduces memory and computational overhead), naturally introduce randomness or can be modified to incorporate randomness without fundamentally altering the training process. However, this kind of structured randomness interacts with data in complex ways, making the associated privacy gains difficult to quantify.

The idea of leveraging randomness for privacy amplification—to reduce the amount of noise required—has been previously explored, primarily in two specific settings. The first is random subsampling, where each data point (or client in federated learning (Kairouz et al., 2021)) is randomly selected to participate in each training iteration (independently across iterations) (Kasiviswanathan et al., 2011; Li et al., 2012; Bassily et al., 2014; Wang et al., 2015; Balle et al., 2018; Wang et al., 2019; Zhu & Wang, 2019; Balle et al., 2020). The second is shuffling, where client data is individually privatized and then randomly reordered to amplify privacy (Erlingsson et al., 2019; Girgis et al., 2021b;a; Feldman et al., 2022; 2023; Chua et al.). More recently, random gradient compression has also been shown to provide privacy amplification (Chen et al., 2024b;a). However, amplification by model parallelism techniques has not been studied before.

In this paper, we investigate privacy amplification in a more general context: whenever each sample (or client in the FL setting) contributes to only a randomly selected portion of the training process, this randomness—if kept secret—can be leveraged to enhance privacy. This includes data partitioning, where a sample (or user) participates in only a subset of the training iterations, or model partitioning, where a sam-

[1]Department of Electrical Engineering, Stanford University, California, USA [2]Microsoft. Correspondence to: Andy Dong <dxa@stanford.edu>.

*Proceedings of the $42^{nd}$ International Conference on Machine Learning*, Vancouver, Canada. PMLR 267, 2025. Copyright 2025 by the author(s).

ple contributes to updating only a randomly chosen subset of model parameters in each iteration. The randomness we leverage in both cases is more structured than independent subsampling or uniform shuffling and has complex interactions with the samples. For example, gradients computed for random subnetworks cannot be represented as simply masked versions of full gradients, so prior privacy amplification methods fail under such scenario.

**Our Contributions:** We develop a unified mathematical framework for privacy amplification that applies to both data and model partitioning. Specifically, our theoretical results establish privacy amplification guarantees in the below settings.

**Model Parallelism:** When a model is partitioned into (disjoint or overlapping) subnetworks and each sample (or client) is assigned to update only a randomly chosen subnetwork in each iteration, this assignment introduces additional privacy beyond standard methods. Existing privacy accounting techniques fail to capture this amplification effect. This privacy gain is particularly relevant for federated and distributed learning, where model parallelism is already employed for computational and memory efficiency, e.g., using various model partitioning techniques (Yuan et al., 2022; Dun et al., 2022; Fang et al., 2024), dropout layers (Hinton et al., 2012; Konečnỳ, 2016) or training models in ensemble configurations (Sagi & Rokach, 2018; Ganaie et al., 2022). Our experiments show that when training a 44 million parameter model with partial model splitting and DP-SGD with $(8, 10^{-5})$-DP, accounting for privacy amplification with our theory yields higher validation accuracy than using existing accounting methods.

**Balanced Iteration Subsampling:** Our framework also extends privacy amplification beyond standard independent subsampling across iterations. We introduce a new subsampling method in which each sample is used in a fixed number of training iterations, rather than being sampled independently in each iteration (as in Poisson Subsampling)[1]. While Poisson subsampling has been a standard tool in DP due to its analytical simplicity, its practical limitations—such as uneven user participation and unpredictable system load—can undermine fair use and operational stability, so has spurred interest in alternative subsampling schemes (Chua et al.). In contrast, Balanced Iteration Subsampling ensures equitable contribution and stable system behavior. Our analysis shows that this method retains the desirable privacy-utility tradeoffs of Poisson sampling while addressing its practical drawbacks, further expanding the toolkit of privacy-preserving subsampling techniques. By rigorously quantifying its privacy guarantees, we offer a compelling alternative to Poisson subsampling, reinforcing the broader insight that

privacy amplification can be achieved through a variety of deployment-friendly approaches.

Balanced Iteration Subsampling can be used in both centralized training and federated learning. In the latter case, it can be viewed as a generalization of random check-ins (Balle et al., 2020), where each client in random check-ins chooses to participate in exactly one iteration with some fixed probability, whereas in our approach, each client participates in exactly $k$ randomly chosen iterations out of $T$ total iterations.

## 2. Background and Definitions

In this section, we provide background and definitions for the paper and set up the problem.

Differenital privacy measures the stability of a randomized algorithm given changes in an input instance, thereby quantifying the extent to which an adversary can infer specific inputs based on the algorithm's output. Mathematically, let $\mathcal{S}$ be the set of all possible datasets. We say that $S, S' \in \mathcal{S}$ are adjacent datasets if $S = S' \cup \{x\}$ or $S' = S \cup \{x\}$ for some single data point $x$.

**Definition 2.1.** (Dwork, 2006) A randomized mechanism $\mathcal{M} : \mathcal{S} \to \Omega$ is said to satisfy $(\epsilon, \delta)$-DP if, for all pairs of adjacent datasets $S, S' \in \mathcal{S}$ and for all measurable set $A$, it holds that $\mathbb{P}(\mathcal{M}(S) \in A) \leq e^\epsilon \mathbb{P}(\mathcal{M}(S') \in A) + \delta$.

It is often useful to analyze privacy guarantees under the notion of Rényi DP (RDP):

**Definition 2.2** (Mironov (2017); Abadi et al. (2016))**.** A randomized mechanism $\mathcal{M} : \mathcal{S} \to \Omega$ is said to satisfy $(\alpha, \epsilon)$-RDP with $\alpha \in (1, \infty)$ if for all pairs of adjacent datasets $S, S' \in \mathcal{S}$, it holds that

$$D_\alpha\left(\mathcal{M}(S) \| \mathcal{M}(S')\right) := \frac{1}{\alpha-1} \log \int_{\omega \in \Omega} \frac{p_{\mathcal{M}(S)}(\omega)^\alpha}{p_{\mathcal{M}(S')}(\omega)^{\alpha-1}} d\omega \leq \epsilon.$$

One key property of RDP is its additive nature under adaptive composition (Mironov, 2017). If mechanisms $\mathcal{M}_1, \mathcal{M}_2, \ldots, \mathcal{M}_k$ each satisfies $(\alpha, \epsilon_i)$-RDP, their combined mechanism satisfies $(\alpha, \sum_{i=1}^k \epsilon_i)$-RDP, simplifying privacy accounting in iterative algorithms and giving a tight composition method.

An $(\alpha, \epsilon(\alpha))$-RDP guarantee can be converted to an $(\epsilon, \delta)$-DP guarantee (Bun & Steinke, 2016; Canonne et al., 2020; Asoodeh et al., 2020):

**Proposition 2.3.** *If $\mathcal{M}$ satisfies $(\alpha, \epsilon(\alpha))$-RDP, it also satisfies $(\epsilon + \frac{\log \frac{1}{\delta}}{\alpha-1}, \delta)$-DP for any $0 < \delta < 1$.*

In this work, we use RDP to analyze the privacy guarantees of model and data partitioning techniques, allowing for fine-grained tracking of privacy loss and facilitating efficient composition analysis in complex training settings.

---

[1]Note that this differs from shuffling, which ensures that each sample appears only once during training.

# 3. Main Results

## 3.1. Main Technical Result

In this section, we present the main result as a mathematical statement outside of the context of differential privacy, although we give an overview of why it is what we need in Remark 3.3. We discuss the intermediate lemmas in the proof of the theorem in a top-down fashion in Section 3.4 and defer the details to the appendix. In Section 3.2, we will show how our main theoretical result applies to quantifying the privacy amplification of model parallelism. In Section 3.3, we will define Balanced Iteration Subsampling and show how the same result applies to quantifying its privacy gain.

**Theorem 3.1.** *Let $S_{d,k} = \{\mu \in \{0,1\}^d \mid \|\mu\|_1 = k\}$ be the set of all binary vectors in $\mathbb{R}^d$ with $k$ number of $1$s and $d - k$ number of $0$s. Let $P = \frac{1}{|S_{d,k}|} \sum_{\mu \in S_{d,k}} \mathcal{N}(c\mu, \sigma^2 I_d)$ be a mixture of Gaussians and $Q = \mathcal{N}(\mathbf{0}, \sigma^2 I_d)$ be a Gaussian centered at $\mathbf{0}$, where $c$ and $\sigma^2$ are positive constants. Then,*

$$\epsilon := \max\{D_\alpha(P \parallel Q), D_\alpha(Q \parallel P)\} \tag{1}$$

$$\leq \max\left\{ \frac{1}{\alpha-1}\log \sum_{\substack{I \in [\binom{d}{k}]^\alpha}} \frac{1}{\binom{d}{k}^\alpha} \exp\left( \frac{c^2}{2\sigma^2} \sum_{\substack{i,j \in [\alpha] \\ i \neq j}} \mu_{I_i}^\mathsf{T} \mu_{I_j} \right), \right.$$

$$\left. \frac{\alpha c^2 k^2}{2\sigma^2 d} + \frac{1}{2(\alpha-1)} \log \frac{\exp\left( \frac{\alpha c^2 k(d-k)}{\sigma^2 d} \right)}{\left( \alpha \exp\left( \frac{c^2 k(d-k)}{\sigma^2 d^2} \right) + (1-\alpha) \right)^d} \right\} \tag{2}$$

$$\leq \max\left\{ \log\left( \frac{1}{\binom{d}{k}} \sum_{l=0}^{k} \binom{k}{l}\binom{d-k}{k-l} \exp\left( \frac{\alpha c^2 l}{2\sigma^2} \right) \right), \right.$$

$$\left. \frac{\alpha c^2 k^2}{2\sigma^2 d} + \frac{1}{2(\alpha-1)} \log \frac{\exp\left( \frac{\alpha c^2 k(d-k)}{\sigma^2 d} \right)}{\left( \alpha \exp\left( \frac{c^2 k(d-k)}{\sigma^2 d^2} \right) + (1-\alpha) \right)^d} \right\} \tag{3}$$

*for all $\alpha \geq 2, \alpha \in \mathbb{N}$, where $[\binom{d}{k}]^\alpha$ is the set of all $\alpha$-length tuples with entries from $\{1, \ldots, \binom{d}{k}\}$ and $\mu_i$ is the $i$th element of $S_{d,k}$.*

*Remark 3.2.* In the equation above, the first term of (1) is upper bounded by the first term of (2), which is in turn upper bounded by the first term of (3); likewise for the second terms. Since the Rényi divergence is not symmetric in its arguments, they are not the same and both are necessary for the completeness of analysis, but numerically the second terms are dominated by the first terms in the max. Also, we note that (2) gives a tighter upper bound while (3) is much more computationally efficient, so one should use (2) whenever possible. We discuss this in more detail in later subsections.

*Remark 3.3.* Although later sections will show applications of Theorem 3.1 and prove why it is useful in the appendix, we preliminarily give an overview of why this is the object

we study in disjoint model splitting where $k = 1$. In model splitting, each $\mu \in S_{d,k}$ corresponds to a potential gradient vector, where an entry of 1 in $\mu$ corresponds to a block whose norm is constrained to $c$ (the gradient clipping norm), while an entry of 0 in $\mu$ corresponds to a block whose gradient is 0 (i.e. not included in the submodel). The set $S_{d,k}$ is then the set of possible gradient vectors generated by the different submodels we could form, before adding noise. On the other hand, the mean of $Q$ is the gradient of $x$ if $x$ is not used in training—which is 0.

The above bounds look complicated and not easily interpretable. We argue that this is intrinsic to the problem of characterizing the Rényi divergence between a mixture of Gaussian and a Gaussian, which is a a difficult problem to tightly characterize with existing results (Michalowicz et al., 2008; Durrieu et al., 2012; Nielsen & Sun, 2016a;b), which do not yield useful bounds for our purposes. The bound is designed to be implemented on a computer to get a numerical answer. In later sections, we will plot this bound against existing baselines when we plug in suitable numbers for a variety of training settings.

## 3.2. Privacy Amplification by Model Parallelism

In this section, we apply Theorem 3.1 to various model parallelism techniques starting with disjoint model splitting. This approach has been utilized in (Yuan et al., 2022; Fang et al., 2024) to limit the memory and computational overhead per node in a distributed setting. We describe this framework in the language of federated learning where each client is assigned a random submodel, while the same discussion also applies to distributed training. In the following sections we extend model parallelism to overlapping subnetworks, where only a subset of the parameters of the model are split between subnetworks (Dun et al., 2022). Finally, we discuss how the analysis applies to models with dropout layers.

### 3.2.1. DISJOINT MODEL SPLITTING

Consider a model with $m$ trainable parameters. In each iteration, we partition the model into $d$ disjoint submodels (which do not need to contain the same number of parameters). The partition may or may not be the same across iterations. That is, we may form completely different submodels in each iteration, or may use the same partitioning strategy throughout. The model partitioning, random or not, is assumed to be *known* by the adversary and is not used for privacy amplification. What we leverage for privacy amplification is the random assignment of one of the $d$ submodels to each client, which remains hidden to an adversary. More precisely, for each client we independently and uniformly select one of the $d$ submodels and send it for an update. The clients compute the gradients on their corresponding submodels and send these gradients back to

**Algorithm 1** Differentially Private Model-Parallel Training

**Input:** $T$ iterations, $m$ model weights, $d$ submodels, dataset $S$, clipping norm $c$, noise variance $\sigma^2$

$M \leftarrow \texttt{init}(m)$    // initialize weights
**for** $t = 1$ **to** $T$ **do**
    $M_1, M_2, \ldots, M_d \leftarrow \texttt{create\_subnets}(M, d)$
    $\texttt{grad\_sum} \leftarrow \texttt{zero\_vector(size}=m)$
    **for** $x$ **in** $S$ **do**
        $k \leftarrow \texttt{random\_integer}(1, 2, \ldots, d)$
        $\texttt{grad} \leftarrow \frac{\partial}{\partial M}\texttt{Loss}(M_k, x)$    // gradient descent, treating gradient of entries not in $M_k$ as 0
        $\texttt{grad} \leftarrow \texttt{clip(grad, }c)$
        $\texttt{grad\_sum} \leftarrow \texttt{grad\_sum} + \texttt{grad}$
    **end for**
    $\texttt{noisy\_grad\_sum} \leftarrow \mathcal{N}(\texttt{grad\_sum, } \sigma^2)$
    $\texttt{update\_model}(M, \texttt{noisy\_grad\_sum})$
**end for**

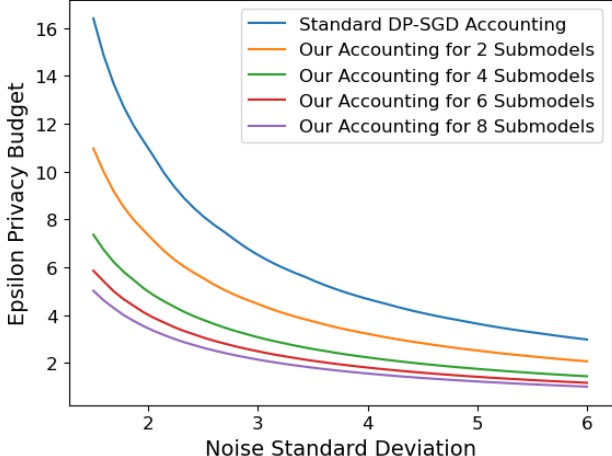

*Figure 1.* The graph compares the noise standard deviation vs the $\epsilon$ privacy cost in $(\epsilon, \delta)$-DP with and without applying our amplification, where $\delta = 10^{-5}$ and we train for 1200 iterations. The data is Poisson subsampled with rate 0.1. Lower $\epsilon$ and less noise are better.

the server. The server collects the gradients, treating the gradient of any parameter not in the submodel as 0, and clips the per-client gradient norm to $c$. The server then sums or averages the clipped gradients and adds per-coordinate Gaussian noise with variance determined through privacy accounting. We summarize the centralized counterpart of this in Algorithm 1, which can be extended to the federated setting by treating the dataset as the set of clients.

**Theorem 3.4.** *Each iteration of Algorithm 1 ($d$ submodels, noise standard deviation $\sigma$, gradient clipping norm $c$) with a deterministic disjoint model splitting method satisfies $(\alpha, \epsilon)$-RDP with $\epsilon$ given by Theorem 3.1, where $k = 1$.*

The proof is in Appendix A.9.

Figure 1 compares the DP guarantee provided by Theorem 3.4 to a baseline analysis with no amplification, for $\{2, 4, 6, 8\}$ submodels. The baseline is calculated using RDP accounting for the DP-SGD Gaussian mechanism (and is shared by all numbers of submodels because existing methods do not take model parallelism into account). The improvement is significant. More figures can be found in the appendix, in Figures 5 and 6.

### 3.2.2. PARTIAL MODEL SPLITTING

In practice, the model splitting method we use may not split the entirety of trainable parameters into disjoint submodels. For example, (Dun et al., 2022) argues that when applying independent subnet training to ResNet, the first few blocks and the last few blocks of the model are sensitive to pruning while the middle blocks are less sentitive. When forming submodels, we may want to include the first and last blocks in all submodels while partitioning the middle blocks into $d$ disjoint sets.

In this such cases, Theorem 3.4 does not directly apply, but

we can proceed by dividing the model into a "non-split" part and a "split" part. In the example above, the non-split part would be the first and the last blocks of ResNet and the split part would be the middle blocks. We can use separate clipping norms for these two parts, and for the non-split part we can use a regular privacy accounting method while the split part can benefit from privacy amplification via the method described in Sections 3.2.1. The RDP costs of the two parts can be directly added together by the composition theorem of RDP.

### 3.2.3. MODEL SPLITTING WITH DROPOUT

In Sections 3.2.1 and 3.2.2, we show how the analysis applies to models with some parameters subject to splitting and some not, while requiring that the parameters subject to splitting be split into disjoint subsets. This section shows that the requirement for disjoint subnets can be relaxed. For example, subnetworks can be created locally and independently at each client by using dropout. This approach has been used in (Konečnỳ, 2016; Wen et al., 2022; Guliani et al., 2022) to limit per client communication and computation.

In Section 3.2.1, any arbitrary way to split the model into disjoint submodels is valid. In fact, if we have multiple ways to split the model into disjoint submodels, we can also use a probabilistic mixture of these disjoint partitionings as stated in the next theorem. In order to assign a subnetwork to a client in a given iteration, we now proceed in two steps. We first choose a way to partition the model into disjoint submodels (possibly in a probabilistic manner, not necessarily uniformly at random) and then one submodel from the chosen partitioning uniformly at random, which is then as-

signed to the client. While the subnetwork assigned to each client comes from this probabilistic mixture of disjoint partitionings, assignment is still independent across different users. Also note that with this probabilistic mixture model, it is possible the submodels trained in a given iteration are no longer disjoint. In the next theorem, we show that a similar privacy guarantee to Theorem 3.4 still holds in this case.

**Theorem 3.5.** *Each iteration of Algorithm 1 (d submodels, noise standard deviation $\sigma$, gradient clipping norm c) with a model splitting method that is a probabilistic mixture of disjoint model splitting methods satisfies $(\alpha, \epsilon)$-RDP with $\epsilon$ given by Theorem 3.1, where $k = 1$.*

The proof is in Appendix A.10.

*Remark* 3.6. We believe Theorem 3.5 can be potentially tightened. It only utilizes the randomness in submodel selection conditioned on disjoint partitioning of the model, but not the randomness in how the disjoint partitioning is selected. Thus, it has the same privacy guarantee as Theorem 3.4.

As a corollary, Theorem 3.5 applies to a special case of dropout—that is, dropout layers that drop each node in the layer (and its associated inputs and outputs) with probability 0.5, which happens to be a common choice of dropout rate when applied to a hidden layer. In a broad sense, dropout can be seen as a form of implicit model parallelism, as it effectively trains different submodels by randomly deactivating neurons during training.

To state it formally, we first note that if a node in a network is dropped out, gradients of the weights touching that node in the preceding and succeeding layers are 0, because in the preceding layer, weights that contribute to the dropped node no longer participate in training, and in the succeeding layer, weights that interact with the dropped node are no longer in training. Assuming dropout is applied to a subset of the hidden layers, we consider the "split part" of the network (see Section 3.2.2) to be the model coefficients preceding and succeeding these dropout layers (this is the set of parameters that can be potentially impacted by dropping the nodes in these layers). Similar to Section 3.2.2, we use Algorithm 2 to update the gradients in this "split part" (and regular DP-SGD on the "non-split part") which results in the following privacy amplification guarantee.

**Corollary 3.7.** *Each iteration of Algorithm 2 (noise standard deviation $\sigma$, gradient clipping norm c) satisfies $(\alpha, \epsilon)$-RDP with $\epsilon$ given by Theorem 3.1, where $d = 2$ and $k = 1$.*

The proof is in Appendix A.11.

---

**Algorithm 2** Differentially Private Training with Dropout

**Input:** $T$ iterations, $m$ model weights, dataset $S$, clipping norm $c$, noise variance $\sigma^2$
$M \leftarrow \texttt{init}(m)$    // initialize weights
**for** $t = 1$ **to** $T$ **do**
  $\texttt{grad\_sum} \leftarrow \texttt{zero\_vector(size=}m\texttt{)}$
  **for** $x$ in $S$ **do**
    $\texttt{grad} \leftarrow \frac{\partial}{\partial M}\text{Loss}(M, x, \texttt{dropout\_rate=0.5})$
    // gradient descent
    $\texttt{grad} \leftarrow \texttt{clip(grad, }c\texttt{)}$
    $\texttt{grad\_sum} \leftarrow \texttt{grad\_sum + grad}$
  **end for**
  $\texttt{noisy\_grad\_sum} \leftarrow \mathcal{N}(\texttt{grad\_sum, } \sigma^2)$
  $\texttt{update\_model}(M\texttt{, noisy\_grad\_sum})$
**end for**

---

### 3.3. Balanced Iteration Subsampling

In this section, we present a novel data subsampling scheme, and show that Theorem 3.1 can be applied to analyze its associated privacy gain. This data subsampling method can be used both in centralized (single-server) private training or in a federated scenario. We note that this data partitioning approach is orthogonal to the model partitioning approaches we studied in the previous section.

Consider the following sampling scheme for differentially private neural network training. Let $T$ be the total number of iterations in the training process. For each sample $x$ in our dataset, we randomly choose $k$ of the $T$ iterations, include $x$ in the $k$ chosen iterations, and not include $x$ in the remaining $T - k$ iterations. Then, for each iteration, we form our training subset by gathering all samples for which we chose this iteration. The randomization in the procedure (i.e. samples used in each iteration) shall remain hidden from the adversary.[2] In the case of federated learning (where instead of samples we assign clients to iterations), this procedure is similar to random check-ins (Balle et al., 2020), in which each client "opts in" at a random iteration with probability $p$ and "opts out" from training entirely with probability $1 - p$. When $p = 1$, this model corresponds to Balanced Iteration Subsampling for $k = 1$.

It is also worthwhile to compare Balanced Iteration Subsampling to Poisson Subsampling (Zhu & Wang, 2019), where in each iteration, we include each sample with probability $\gamma$. The main difference is that in Poisson sampling, the number of times each sample is included in training follows a Binomial$(T, \gamma)$ distribution, with an expectation of $T\gamma$ but it could range from 0 to $T$, whereas in our case, each sample is included in training exactly $k$ times. If $\gamma = \frac{k}{T}$, as $T \to \infty$,

---

[2]A concurrent work (Feldman & Shenfeld, 2025) analyzes the same subsampling technique and proposes an RDP bound that is a special case of our (2) with $k = 1$. See Remark 3.9 for more.

these two subsampling schemes become equivalent.

**Theorem 3.8.** *Balanced Iteration Subsampling composed with Gaussian Mechanism with $l_2$ sensitivity $c$ satisfies $(\alpha, \epsilon)$-RDP with $\epsilon$ given by Theorem 3.1, where $d = T$, the total number of iterations, and $k$ is the number of iterations each sample participates in.*

The proof is found in Appendix A.12.

*Remark* 3.9. When $k = 1$, the first term in (2) is exactly the same as the expression computed in the unpublished work (Liew & Takahashi, 2022) and the concurrent work (Feldman & Shenfeld, 2025) modulo minor combinatorial simplifications, where it appears that in the former, the authors mistakenly use the bound for the Shuffle Gaussian Mechanism. In the Shuffle Gaussian Mechanism, each sample is used in only one random iteration, but the assignments of different samples to iterations are dependent since the batch size used in each iteration is fixed, so the analysis (A.12) does not apply to this case. Our Balanced Iteration Subsampling with $k = 1$ is similar to the Shuffle Gaussian Mechanism in the sense that each sample is used in only one random iteration but it maintains independence across samples. We also remark that the second term in (2) is numerically smaller than the first term, so (2) and the aforementioned works are practically the same, while (3) is slightly looser but much more computationally efficient.

*Remark* 3.10. When applying Balanced Iteration Subsampling, we could also use a small $T$ and $k$ and compose $N$ sequential training runs, where each run would be one epoch. Doing so would incur a slightly higher privacy cost than running $TN$ iterations total and using each data point in $kN$ random iterations, which is intuitive because there is less randomness in the first case when the iterations are organized in smaller epochs. Figure 4 showcases this.

It is natural to compare the privacy guarantee in Theorem 3.8 to Poisson Subsampling, whose RDP privacy guarantee is stated in the below theorem for reference.

**Theorem 3.11.** *(Zhu & Wang, 2019) The Poisson Subsampled Gaussian Mechanism with $l_2$ sensitivity $c$ and subsampling rate $\gamma$ satisfies $(\alpha, \epsilon)$-RDP with*

$$\epsilon(\alpha) = \frac{1}{\alpha - 1} \log \left\{ (1 - \gamma)^{\alpha - 1} (\alpha \gamma - \gamma + 1) \right.$$
$$\left. + \sum_{l=2}^{\alpha} \binom{\alpha}{l} (1 - \gamma)^{\alpha - l} \gamma^l \exp\left(\frac{l(l-1)c^2}{2\sigma^2}\right) \right\}.$$

When using the tighter upper bound given by (2), the RDP privacy guarantee of Balanced Iteration Subsampling is always better than Poisson Subsampling, with a small margin when $T, k$ are large and a slightly larger margin when $T$ is very small. The computationally efficient upper bound (3)

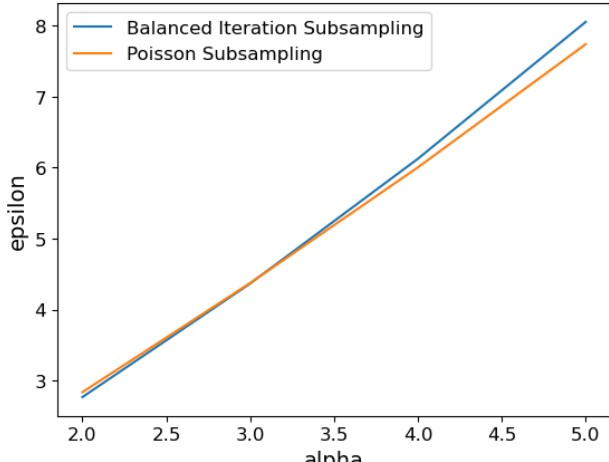

*Figure 2.* The $\epsilon(\alpha)$ vs $\alpha$ graph for Balanced Iteration Subsampling and Poisson Subsampling, where $T = 1000$, $k = 100$, and $\sigma = 2$. Lower $\epsilon$ is better. For a large $T$ like this, the noise standard deviation to achieve the same privacy guarantee is almost identical.

is better than Poisson Subsampling whenever $\frac{k}{T} \gtrsim 0.2$ and similarly wins by a larger margin when $T$ is very small.

The similarity in performance of these two subsampling methods when $T, k$ are large is intuitive, because the number of times a data point gets used by Poisson Subsampling tends to be close to its expectation as the number of iterations is large, so the two methods become similar in the limit $T \to \infty$. Figure 2 shows this.

On the other hand, when $T$ is small—for example, on the order of 10—there is a nontrivial probability that the number of times Poisson Subsampling picks a sample significantly exceeds the expectation, which will leak more information about that particular sample. In these cases, a large $\alpha$ will keenly capture this nontrivial probability of failure, leading to a sharp increase in the RDP privacy guarantee $\epsilon$ we can provide for Poisson Subsampling, as illustrated in Figure 3. This comparison also sheds light on why the $\epsilon_{\text{Poisson}}(\alpha)$ curve spikes up for large values of $\alpha$ relative to $T$. Figure 4 showcases the difference between the two subsampling methods when converted to the $(\epsilon, \delta)$-DP domain.

More comparisons of these two subsampling methods (both the $T$ large and $T$ small cases) can be found in the Appendix in Figures 7 and 8.

In conclusion, we presented Balanced Iteration Subsampling as an appealing alternative to Poisson Subsampling with better privacy guarantees. More importantly, we have shown that two completely different sources of randomness in model and data partitioning can have a similar structure underneath and can yield a nontrivial privacy amplification

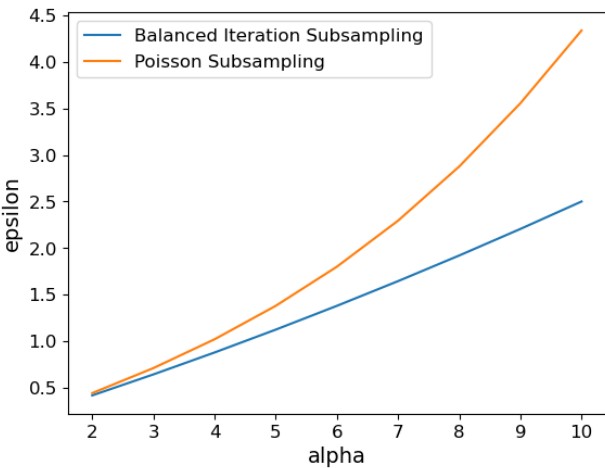

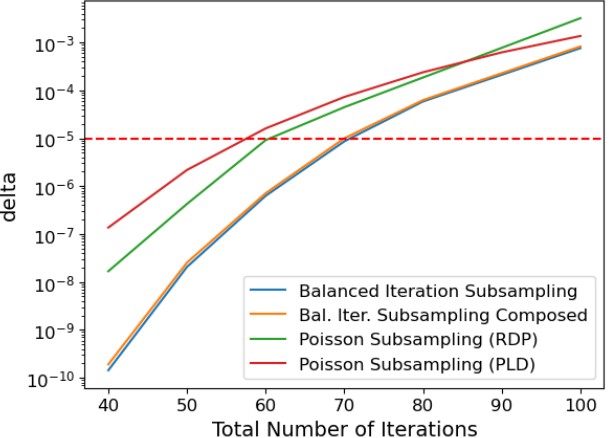

*Figure 3.* The $\epsilon(\alpha)$ vs $\alpha$ graph for Balanced Iteration Subsampling and Poisson Subsampling, where $T = 10$, $k = 4$, and $\sigma = 2$. Lower $\epsilon$ is better. As $\alpha$ becomes large, Balanced Iteration Subsampling starts to perform much better than Poisson Subsampling.

*Figure 4.* The $\delta$ vs $T$ graph for Balanced Iteration Subsampling and Poisson Subsampling, where $\epsilon = 10$, $\gamma = \frac{k}{T} = 0.4$, and $\sigma = 2$. Lower $\delta$ and more iterations are better. Balanced Iteration Subsampling Composed treats 10 iterations as an epoch, using each sample 4 times in each epoch, and composing multiple epochs together, as suggested in Remark 3.10. Poisson Subsampling accounting is done using RDP with Theorem 3.11 and using PLD with Google's PLD library (Google Differential Privacy Team, 2024). Under these settings, with a commonly used budget $\delta = 10^{-5}$ (the dotted line), either method of Balanced Iteration Subsampling is able to perform 10 more iterations out of 60.

gain via the same analysis.

### 3.4. Proof Overview of Theorem 3.1

In this final subsection, we present an overview of the proof of Theorem 3.1.

We first bound $D_\alpha(P\|Q)$—which we refer to as the forward divergence—with the help of the following lemmas.

**Lemma 3.12.** *Let* $P = \sum_{i=1}^{n} \nu_i \mathcal{N}\left(\mu_i, \sigma^2 I\right)$ *and* $Q = \mathcal{N}\left(0, \sigma^2 I\right)$ *for some vectors* $\mu_i$, $1 \le i \le n$, *and probability measure* $\nu$. *Then,*

$$D_\alpha(P \parallel Q)$$

$$= \frac{1}{\alpha - 1} \log \sum_{I \in [n]^\alpha} \left( \prod_{i \in I} \nu_i \right) \exp\left( \frac{1}{2\sigma^2} \sum_{\substack{i,j \in [\alpha] \\ i \ne j}} \mu_{I_i}^{\mathsf{T}} \mu_{I_j} \right).$$

The proof is in Appendix A.1. If we apply this to the setting in Theorem 3.1, it gives the first item in Eq. (2). Although this gives a non-integral representation of $D_\alpha(P \parallel Q)$, the time complexity to compute the RHS is still $O(n^\alpha)$, where $n$ is the number of mixtures in $P$, and in this case, $\binom{d}{k}$, so the time complexity to compute the RHS is $O\left(d^{k\alpha}\right)$. In practice, to do privacy accounting, we need to efficiently calculate Rényi DP for $\alpha = 2, 3, \ldots, 100$, so we need the following lemma to further reduce computation.

**Lemma 3.13.** *Let* $P_i = \mathcal{N}(\mu_i, \sigma^2 I)$ *for* $i = 1, 2, \ldots, n$ *be the elements of a mixture of Gaussians. For a mixture measure* $\nu$, *if for two randomly chosen mixture centers*

$\mu_j, \mu_k \overset{i.i.d.}{\sim} \nu$ *it holds that* $\mu_j^{\mathsf{T}} \mu_k \overset{dist.}{=} \mu_j^{\mathsf{T}} \mu_k \mid \mu_j$, *then*

$$D_\alpha \left( \sum_{i=1}^{n} \nu_i \mathcal{N}\left(\mu_i, \sigma^2 I\right) \, \middle\| \, \mathcal{N}\left(0, \sigma^2 I\right) \right)$$

$$\le D_2 \left( \sum_{i=1}^{n} \nu_i \mathcal{N}\left(\mu_i, \frac{2\sigma^2}{\alpha} I\right) \, \middle\| \, \mathcal{N}\left(0, \frac{2\sigma^2}{\alpha} I\right) \right)$$

*for* $\alpha \ge 2, \alpha \in \mathbb{N}$.

The proof is in Appendix A.2. The mixture centers $S_{d,k}$ and mixture measure that assigns $\frac{1}{|S_{d,k}|}$ to everything in Theorem 3.1 satisfy the requirement of Lemma 3.13 because $S_{d,k}$ contains all permutations of the vector comprised of $k$ 1s and $d - k$ 0s, so by symmetry, under $\mu_j, \mu_k \overset{i.i.d.}{\sim} \nu$, the distribution of $\mu_j^{\mathsf{T}} \mu_k$ is invariant given $\mu_j$. Combining Lemma 3.13 with Lemma 3.12 and plugging in $\alpha = 2$, we get the following upper bound on the forward divergence.

**Corollary 3.14.** *Let P and Q be defined the same way as in Theorem 3.1. Then,*

$$D_\alpha(P \parallel Q) \le \log \left( \frac{1}{\binom{d}{k}} \sum_{l=0}^{k} \binom{k}{l} \binom{d-k}{k-l} \exp\left( \frac{\alpha c^2 l}{2\sigma^2} \right) \right).$$

*When $k = 1$, the above is further simplified to*

$$\log\left(\frac{1}{d}\left(\exp\left(\frac{\alpha c^2}{2\sigma^2}\right) + (d-1)\right)\right).$$

Corollary 3.14 merges the summation of $\binom{d}{k}^2$ terms into a sum of combinatorial terms. A detailed breakdown is in Appendix A.3. It can now be computed in $O(dk)$ time when $k > 1$ and $O(1)$ time when $k = 1$.

The following lemmas build up the upper bound of $D_\alpha(Q \parallel P)$—which we refer to as the reverse divergence.

**Lemma 3.15.** *Let $S_{d,k} = \{\mu \in \{0,1\}^d \mid \|\mu\|_1 = k\}$ be the set of all binary vectors in $\mathbb{R}^d$ with $k$ number of 1s and $d - k$ number of 0s. Let $P = \frac{1}{|S_{d,k}|}\sum_{\mu \in S_{d,k}} \mathcal{N}(c\mu, \sigma^2 I)$ and $Q = \mathcal{N}(\mathbf{0}, \sigma^2 I)$, where $c$ and $\sigma^2$ are constants. Let $Q' = \mathcal{N}(\frac{c}{|S_{d,k}|}\sum_{\mu \in S_{d,k}}\mu,\ \sigma^2 I) = \mathcal{N}\left(\frac{ck}{d}\mathbf{1}, \sigma^2 I\right)$ be a Gaussian centered at the average of P's mixture centers. Then*

$$D_\alpha(Q \parallel P) = D_\alpha(Q \parallel Q') + D_\alpha(Q' \parallel P)$$

*for all $\alpha > 1$.*

*Remark* 3.16. The setup in the lemma can be more generalized, but the current statement is sufficient for our purpose.

Again, the proof is deferred to Appendix A.4. We now seek to calculate or upper bound each of $D_\alpha(Q \parallel Q')$ and $D_\alpha(Q' \parallel P)$ individually. Since $Q$ and $Q'$ are both multivariate Gaussians with the same covariance matrix, we can compute the exact divergence between them.

**Corollary 3.17.** *Let $Q$ and $Q'$ be defined the same way as in Lemma 3.15. We have*

$$D_\alpha(Q \parallel Q') = \frac{\alpha c^2 k^2}{2\sigma^2 d}.$$

A proof is in Appendix A.6.

The following lemma deals with the second addend on the RHS of Lemma 3.15.

**Lemma 3.18.** *Let $S_{d,k} = \{\mu \in \{0,1\}^d \mid \|\mu\|_1 = k\}$ be the set of all binary vectors in $\mathbb{R}^d$ with $k$ number of 1s and $d - k$ number of 0s. For some fixed constants $c, \sigma^2$, let $\mu_0 = \frac{1}{|S_{d,k}|}\sum_{\mu \in S_{d,k}}\mu = \frac{k}{d}$ be the mean of $S_{d,k}$. Let $P'' = \frac{1}{|S_{d,k}|}\sum_{\mu \in S_{d,k}} \mathcal{N}(c(\mu - \mu_0), \sigma^2 I_d)$ be a mixture of Gaussians with centers $S_{d,k}$ normalized to have a mean of $\mathbf{0}$ and scaled by c. Let $Q'' = \mathcal{N}(\mathbf{0}, \sigma^2 I_d)$. Then,*

$$D_\alpha(Q'' \parallel P'') \le D_\alpha\left(Q'' \,\Big\|\, \mathcal{N}\left(\mathbf{0}, \sigma^2 \exp\left(\frac{c^2 k(d-k)}{\sigma^2 d^2}\right) I_d\right)\right)$$

*for all $\alpha \ge 2$.*

The proof is again deferred to Appendix A.7. Note that $Q''$ and $P''$ in Lemma 3.18 are the same as $Q'$ and $P$ in

Lemma 3.15 but offset by $-\frac{ck}{d}\mathbf{1}$, so we can apply the translational invariance of Renyi divergence.

Now we have two multivariate Gaussians with the same mean but different covariance matrices, whose Rényi divergence we can also calculate exactly.

**Corollary 3.19.** *Let $Q''$ and $P''$ be defined the same way as in Lemma 3.18. Let $Q'$ and $P$ be defiend the same way as in Lemma 3.15. Then,*

$$
\begin{aligned}
&D_\alpha(Q \parallel P') \\
&= D_\alpha(Q'' \parallel P'') \qquad \text{by translational invariance} \\
&\le \frac{1}{2(\alpha-1)} \log \frac{\exp\left(\frac{\alpha c^2 k(d-k)}{\sigma^2 d}\right)}{\left(\alpha \exp\left(\frac{c^2 k(d-k)}{\sigma^2 d^2}\right) + (1-\alpha)\right)^d}.
\end{aligned}
$$

The proof is in Appendix A.8.

*Proof of Theorem 3.1.* Combining Corollary 3.17 and Corollary 3.19 bounds RHS of Lemma 3.15. Further combining with Lemma 3.12 and Corollary 3.14 gives Theorem 3.1. □

# 4. Experiments

## 4.1. Model Splitting

In this section, we train ResNet-101 on CIFAR-10 with model splitting under both centralized setting and federated setting, and analyze their respective privacy guarantees by using the techniques and theoretical results presented in Sections 3.2.1 and 3.2.2. We simulate a differentially private fine-tuning scenario. We split the dataset into two halves, each containing half of the images from each class. We use the first half to pre-train the model without DP, but only with 8 of the 10 classes. This non-private pretraining resembles having access to public data with a different distribution or a pretrained model for a different task. We end the pre-training when validation accuracy reaches 70%. Then, we use the second half of the dataset for private finetuning. Under centralized training, we train for 1000 iterations using $(8, 10^{-5})$-DP. Under federated training, each user holds 2 samples and we train for 250 sessions. In each training session, each user trains their received model locally for 3 iterations, sends the model update to the server, and the server aggregates the model updates and adds noise to ensure a user-level $(8, 10^{-5})$-DP. To make the training compatible with DP, we replace the batch normalization layers in the model with group normalization layers.

We compare three solutions. The first "baseline" uses independent subnet training for private fine-tuning as introduced in Section 3.2.2 but the privacy analysis is done with existing tools (i.e., the standard RDP accounting for DP-SGD pipeline). The second "amplification" uses the same exact

*Table 1.* Comparison of three domain adaptation methods with a data subsampling rate of 0.1, under centralized setting for sample-level $(8, 10^{-5})$-DP. Noise standard deviation is relative to the clipping norm (and already divided by expected batch size). Validation accuracies are best of 3 random seeds and have standard deviations around $0.7\%$.

| METHOD | # OF SUB-MODELS | VALIDATION ACCURACY | NOISE STD. DEV. | ADDRESSES LIMITED COMPUTE? |
|---|---|---|---|---|
| BASELINE | 3 | 79.80% | $6.62 \times 10^{-4}$ | $\checkmark$ |
| AMPLIFICATION (OURS) | 3 | 82.43% | $5.44 \times 10^{-4}$ | $\checkmark$ |
| BASELINE | 8 | 76.80% | $6.62 \times 10^{-4}$ | $\checkmark\checkmark$ |
| AMPLIFICATION (OURS) | 8 | 80.52% | $4.96 \times 10^{-4}$ | $\checkmark\checkmark$ |
| RELAXATION | 1 | 84.99% | $6.62 \times 10^{-4}$ | $\times$ |

*Table 2.* Comparison of three domain adaptation methods under federated setting for user-level $(8, 10^{-5})$-DP. Noise standard deviation is relative to the clipping norm (and already divided by number of users). Validation accuracies are best of 3 random seeds and have standard deviations around $0.7\%$.

| METHOD | # OF SUB-MODELS | VALIDATION ACCURACY | NOISE STD. DEV. | ADDRESSES LIMITED COMPUTE? |
|---|---|---|---|---|
| BASELINE | 3 | 78.47% | $6.75 \times 10^{-4}$ | $\checkmark$ |
| AMPLIFICATION (OURS) | 3 | 80.28% | $5.72 \times 10^{-4}$ | $\checkmark$ |
| BASELINE | 8 | 76.96% | $6.75 \times 10^{-4}$ | $\checkmark\checkmark$ |
| AMPLIFICATION (OURS) | 8 | 79.11% | $5.36 \times 10^{-4}$ | $\checkmark\checkmark$ |
| RELAXATION | 1 | 82.18% | $6.75 \times 10^{-4}$ | $\times$ |

training method but privacy accounting is done with our method in Section 3.2. In both cases, we do not split the first 5 and last 2 blocks of ResNet-101 but partition the middle 22 blocks into three or eight disjoint sets to form subnetworks, one of which is then randomly assigned to each sample. The third "relaxation" assumes client compute is not a problem and each client tunes the full model (without model splitting). The results are shown in Table 1 for centralized setting and Table 2 for federated setting. The results of this experiment show that in both setting, the use of model parallelism techniques to handle limited client compute is a viable strategy with some decrease in overall accuracy due to model splitting, but our privacy amplification technique improves the accuracy of model splitting by allowing to inject less noise for the same privacy guarantee.

## 4.2. Balanced Iteration Subsampling

We experimented with differentially private training using Balanced Iteration Subsampling vs Poisson Subsampling as the data subsampling technique. Since the number of training iterations is large (on the order of 1000), the two methods have a similar training dynamic and privacy guarantee. In particular, the noise standard deviations differ from each other by less than 1% (similar to Figure 8b). Thus, the statistics of the validation accuracy show little difference. For training WideResNet-40-4 from scratch on CIFAR-10, $(8, 10^{-5})$-DP, 2000 iterations, using each sample 655 times for Balanced Iteration Subsampling and with probability $\frac{655}{2000}$ in each iteration for Poisson Subsampling. Balanced Iteration Subsampling injects noise with $\sigma = 10.17$ for validation accuracy of $70.21\%$, while Poisson Subsampling injects noise with $\sigma = 10.20$ for validation accuracy of $70.13\%$. For finetuning ResNet-101 (using the same setting as Section 4.1 but without model splitting), Balanced Iteration Subsampling injects noise with $\sigma = 2.36$ for validation accuracy of $84.86\%$, while Poisson Subsampling injects noise with $\sigma = 2.34$ for validation accuracy of $84.99\%$. The results are best of 3 random seeds. This shows that the two subsampling methods achieve very similar privacy-accuracy tradeoffs experimentally, which reinforces our point that Balanced Iteration Subsampling is a valid alternative when we cannot apply Poisson Subsampling.

## 5. Conclusion

In this work, we identify and leverage the inherent randomness in ML training algorithms for privacy amplification. These sources of randomness emerge whenever each sample interacts with a randomly selected portion of the training process, whether through submodels or structured subsampling. However, existing methods have not fully captured these effects. We introduce a mathematical framework to quantify the privacy gains achieved through model splitting and dropout—two techniques used in federated learning that help reduce computational and communication costs—as well as through Balanced Iteration Subsampling, an alternative to Poisson Subsampling that provides improved privacy guarantees in certain regimes.

This work makes one step forward to systematically quantifying all forms of randomness encountered in training and their contributions to privacy amplification. To this end, we provide a deeper understanding of how structured randomness enhances privacy-preserving machine learning. A more rigorous analysis of these interactions will lead to better theoretical bounds and improved implementations of privacy-preserving methods. A promising future direction is to extend this framework to other training paradigms, further strengthening the connection between structured randomness and differential privacy.

## Acknowledgments

This work was partially supported by the NSF grant CIF-2213223.

Some of the computing for this project was performed on the Sherlock cluster. We would like to thank Stanford University and the Stanford Research Computing Center for providing computational resources and support that contributed to these research results.

## Impact Statement

This work represents an important first step in quantifying the privacy amplification effects of model parallelism methods, offering new insights into enhancing privacy-preserving machine learning. By leveraging the inherent randomness in model partitioning and subsampling, our approach contributes to the development of more secure and efficient distributed learning frameworks, particularly in federated learning settings. This study lays the groundwork for further exploration into the interplay between model parallelism and differential privacy, fostering ongoing innovation in privacy-preserving AI.

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

# A. Proofs.

## A.1. Proof of Lemma 3.12

Let $d$ be the dimension of $\mu_i$. Then,

$$
\begin{aligned}
D_\alpha(P \parallel Q) &= \frac{1}{\alpha - 1} \log \int_{\mathbb{R}^d} P^\alpha Q^{1-\alpha} dx \\
&= \frac{1}{\alpha - 1} \log \int_{\mathbb{R}^d} \frac{1}{(2\pi\sigma^2)^{\frac{d}{2}}} \sum_{I \in [n]^\alpha} \left( \prod_{i \in I} \nu_i \right) \exp\left( -\frac{1}{2\sigma^2} \left( \sum_{i \in I} \|x - \mu_i\|_2 - (\alpha - 1)\|x\|_2 \right) \right) dx \\
&= \frac{1}{\alpha - 1} \log \sum_{I \in [n]^\alpha} \left( \prod_{i \in I} \nu_i \right) \int_{\mathbb{R}^d} \frac{1}{(2\pi\sigma^2)^{\frac{d}{2}}} \exp\left( -\frac{1}{2\sigma^2} \left( \sum_{i \in I} \|x - \mu_i\|_2 - (\alpha - 1)\|x\|_2 \right) \right) dx \\
&= \frac{1}{\alpha - 1} \log \sum_{I \in [n]^\alpha} \left( \prod_{i \in I} \nu_i \right) \exp\left( \frac{1}{2\sigma^2} \sum_{i,j \in [\alpha],\, i \neq j} \mu_{I_i}^\mathsf{T} \mu_{I_j} \right).
\end{aligned}
$$

The $\mathsf{T}$ in the superscript means transpose. The second line comes from expanding the exponentials. The third line comes from linearity. The fourth line comes from completing the squares then integrating multivariate Gaussian densities. $\quad\square$

## A.2. Proof of Lemma 3.13

Apply Lemma 3.12 to both sides and replace $\nu$ with the uniform distribution to get

$$
D_\alpha(P \parallel Q) = \frac{1}{\alpha - 1} \log \left( \frac{1}{n^\alpha} \sum_{I \in [n]^\alpha} \exp\left( \frac{1}{\sigma^2} \sum_{i,j \in [\alpha],\, i < j} \mu_{I_i}^\mathsf{T} \mu_{I_j} \right) \right). \tag{4}
$$

Similarly,

$$
D_2 \left( \sum_{i=1}^n \nu_i \mathcal{N}\left( \mu_i, \frac{2\sigma^2}{\alpha} \right) \,\middle\|\, \mathcal{N}\left( 0, \frac{2\sigma^2}{\alpha} \right) \right) = \log \left( \frac{1}{n^2} \sum_{I \in [n]^2} \exp\left( \frac{1}{\sigma^2} \frac{\alpha}{2} \mu_{I_1}^\mathsf{T} \mu_{I_2} \right) \right).
$$

Exponentiating both sides and rearranging, we want to show that

$$
\frac{1}{n^\alpha} \sum_{I \in [n]^\alpha} \exp\left( \frac{1}{\sigma^2} \sum_{i,j \in [\alpha],\, i < j} \mu_{I_i}^\mathsf{T} \mu_{I_j} \right) \overset{?}{\leq} \left( \frac{1}{n^2} \sum_{I \in [n]^2} \exp\left( \frac{1}{\sigma^2} \frac{\alpha}{2} \mu_{I_1}^\mathsf{T} \mu_{I_2} \right) \right)^{\alpha - 1}.
$$

Notice that the LHS is the MGF $M_X(t)$ of the random variable

$$
X = \sum_{i,j \in [\alpha],\, i < j} \mu_{I_i}^\mathsf{T} \mu_{I_j}
$$

where $I \sim \mathrm{Unif}([n]^\alpha)$ and $t = \frac{1}{\sigma^2}$. The RHS is likewise the MGF $M_Y(t)$ of $Y$, who is the sum of $\alpha - 1$ i.i.d. copies of $\frac{\alpha}{2} \mu_{T_1}^\mathsf{T} \mu_{T_2}$ where $T \sim \mathrm{Unif}([n]^2)$, i.e.,

$$
Y = \frac{\alpha}{2} \sum_{k=1}^{\alpha - 1} \mu_{T_1^k}^\mathsf{T} \mu_{T_2^k}
$$

where $T^k \overset{i.i.d.}{\sim} \mathrm{Unif}\left([n]^2\right)$ for $k = 1, 2, \ldots, \alpha - 1$, and $t = \frac{1}{\sigma^2}$. Next, since each entry of $T^k$ is chosen according to $\nu$ and we assume that $\mu_j^\mathsf{T} \mu_k \overset{dist.}{=} \mu_j^\mathsf{T} \mu_k \mid \mu_j$ for $\mu_j, \mu_k \overset{i.i.d.}{\sim} \nu$, we can set $T_1^k$ to be equal to $T_1^1$ for $k = 2, \ldots, \alpha - 1$ and leave $Y$ with the same distribution (and thus the same MGF). Next, since now the $T_1^k$'s are the same, we can set up a bijection between outcomes of $I$ and the outcomes of $T_k$ as follows:

$$
(I_1, I_2, I_3, I_4, \ldots, I_\alpha) \quad \longleftrightarrow \quad (T_1^1, T_2^1, T_2^2, T_2^2, \ldots, T_2^{\alpha - 1})
$$

and can verify that both random objects follow the same distribution $\mathrm{Unif}([n]^\alpha)$. This means if now we let $I \sim \mathrm{Unif}([n]^\alpha)$ and

$$
X = \sum_{i,j \in [\alpha],\, i<j} \mu_{I_i}^\mathsf{T} \mu_{I_j}, \qquad Y = \frac{\alpha}{2} \sum_{i=2}^{\alpha} \mu_{I_1}^\mathsf{T} \mu_{I_i},
$$

these two random variables would have the same distribution as first specified. Next, we condition on a specific histogram $h$ (where histogram is defined in the above section). The random variable $X \mid h$ is a constant as $X$ sums up all pair-wise correlations no matter the order. On the other hand, the distribution of $Y \mid h$ is non-trivial but has mean equal to $X \mid h$ because we can rewrite $X$ and $Y$ as

$$
Y_j = \sum_{i=1}^{\alpha} \mu_{I_j}^\mathsf{T} \mu_{I_i} - \mu_{I_j}^\mathsf{T} \mu_{I_j}, \qquad X = \frac{1}{2} \sum_{j=1}^{\alpha} Y_j, \qquad Y = \frac{\alpha}{2} Y_1
$$

and the $Y_j$'s have the same distribution given $h$. So, we have

$$
\mathbb{E}[e^{tY} \mid h] \geq e^{t\mathbb{E}[Y|h]} = \mathbb{E}[e^{tX} \mid h] \qquad \text{for } t \geq 0.
$$

Note that $t \geq 0$ is always satisfied as $t = \frac{1}{\sigma^2} \geq 0$. With the Law of Iterated Expectations,

$$
M_Y(t) = \mathbb{E}_h \Big[ \mathbb{E}_I[e^{tY} \mid h] \Big] \geq \mathbb{E}_h \Big[ \mathbb{E}_I[e^{tX} \mid h] \Big] = M_X(t).
$$

$\square$

### A.3. Proof of Corollary 3.14

The mixture centers of $P$ are

$$
c \cdot \left\{ v \in \{0,1\}^d \;\middle|\; \sum_{i=1}^{d} v_i = k \right\}.
$$

Combining Lemma 3.13 and Equation 4, we have the following upper bound on the forward divergence:

$$
D_\alpha(P \parallel Q) \leq \log \left( \frac{1}{\binom{d}{k}^2} \sum_{i=1}^{\binom{d}{k}} \sum_{j=1}^{\binom{d}{k}} \exp\left( \frac{\alpha}{2\sigma^2} \mu_i^\mathsf{T} \mu_j \right) \right).
$$

First, since $S_{d,k}$ contains all permutations of binary vectors with $k$ 1s, by symmetry, the value of $\sum_{j=1}^{n} \exp\left( \frac{\alpha}{2\sigma^2} \mu_i^\mathsf{T} \mu_j \right)$ is not dependent on $\mu_i$, so we reduce it to

$$
\log \left( \frac{1}{\binom{d}{k}} \sum_{j=1}^{n} \exp\left( \frac{\alpha}{2\sigma^2} \mu_1^\mathsf{T} \mu_j \right) \right).
$$

Next, we count how many ways $\mu_1^\mathsf{T} \mu_j$ can equal $c^2 l$ for $0 \leq l \leq k$. The vector $\mu_j$ needs to have $l$ number of 1s in the first $k$ entries, and the rest $k - l$ number of 1s in the last $d - k$ entries, giving a total of $\binom{k}{l}\binom{d-k}{k-l}$ ways. So,

$$
D_\alpha(P \parallel Q) \leq \log \left( \frac{1}{\binom{d}{k}} \sum_{l=0}^{k} \binom{k}{l}\binom{d-k}{k-l} \exp\left( \frac{\alpha c^2 l}{2\sigma^2} \right) \right). \tag{5}
$$

$\square$

In the case that $k = 1$ (used in model splitting), the above is further reduced to

$$
\log \left( \frac{1}{\binom{d}{1}} \left( \binom{1}{0}\binom{d-1}{1} \exp(0) + \binom{1}{1}\binom{d-1}{0} \exp\left( \frac{\alpha c^2}{2\sigma^2} \right) \right) \right) = \log \left( \frac{1}{d} \left( \exp\left( \frac{\alpha c^2}{2\sigma^2} \right) + (d-1) \right) \right).
$$

## A.4. Proof of Lemma 3.15

Without loss of generality, let $c = 1$ as $c$ is a scaling factor for all Gaussian centers so we can instead re-scale the standard deviation $\sigma$ to achieve the same effect. To prove the lemma, we apply an offset vector $-\frac{k\alpha}{d}\mathbf{1}$ to the integrand of $D_\alpha(Q' \| P)$,

$$
D_\alpha(Q' \| P) = \frac{1}{\alpha - 1} \log \left( (2\pi\sigma^2)^{-\frac{d}{2}} \int_{\mathbb{R}^d} \frac{\left[ \prod_{i=1}^d \exp\left( -\frac{1}{2\sigma^2} \left( x_i - \frac{k}{d} \right)^2 \right) \right]^\alpha}{\left[ \frac{1}{|S_{d,k}|} \sum_{\mu \in S_{d,k}} \prod_{i=1}^d \exp\left( -\frac{1}{2\sigma^2}(x_i - \mu_i)^2 \right) \right]^{\alpha-1}} dx_1 \cdots dx_d \right)
$$

$$
= \frac{1}{\alpha - 1} \log \left( (2\pi\sigma^2)^{-\frac{d}{2}} \int_{\mathbb{R}^d} \frac{\left[ \prod_{i=1}^d \exp\left( -\frac{1}{2\sigma^2} \left( x_i - \frac{k}{d} + \frac{k\alpha}{d} \right)^2 \right) \right]^\alpha}{\left[ \frac{1}{|S_{d,k}|} \sum_{\mu \in S_{d,k}} \prod_{i=1}^d \exp\left( -\frac{1}{2\sigma^2} \left( x_i - \mu_i + \frac{k\alpha}{d} \right)^2 \right) \right]^{\alpha-1}} dx_1 \cdots dx_d \right).
$$

In the exponents of the numerator,

$$
\left( x_i - \frac{k}{d} + \frac{k\alpha}{d} \right)^2 = \left( x_i - \frac{k}{d}(1 - \alpha) \right)^2 = x_i^2 + 2\frac{k}{d}(\alpha - 1)x_i + \left( \frac{k}{d}(\alpha - 1) \right)^2.
$$

The numerator can then be re-written as

$$
\left[ \prod_{i=1}^d \exp\left( -\frac{1}{2\sigma^2} x_i^2 \right) \right]^\alpha \cdot \prod_{i=1}^d \exp\left( -\frac{1}{2\sigma^2} \cdot 2\frac{k}{d}\alpha(\alpha - 1)x_i \right) \cdot \exp\left( -\frac{1}{2\sigma^2} \cdot \alpha d \left( \frac{k}{d}(\alpha - 1) \right)^2 \right).
$$

In the exponents of the denominator, $\mu_i$ is either 1 or 0 since $\mu$ is a binary vector. Case by case,

$$
\left( x_i - 1 + \frac{k\alpha}{d} \right)^2 = (x_i - 1)^2 + 2\frac{k\alpha}{d}x_i - 2\frac{k\alpha}{d} + \left( \frac{k\alpha}{d} \right)^2
$$

and

$$
\left( x_i + \frac{k\alpha}{d} \right)^2 = x_i^2 + 2\frac{k\alpha}{d}x_i + \left( \frac{k\alpha}{d} \right)^2.
$$

Since every $\mu$ contains $k$ number of 1s and $d - k$ number of 0s, exactly $k$ of the terms in the product have the former form and the rest $d - k$ terms have the latter form. We can re-write the denominator as

$$
\left[ \frac{1}{|S_{d,k}|} \sum_{\mu \in S_{d,k}} \prod_{i=1}^d \exp\left( -\frac{1}{2\sigma^2}(x_i - \mu_i)^2 \right) \right]^{\alpha-1} \cdot \prod_{i=1}^d \exp\left( -\frac{1}{2\sigma^2} \cdot 2\frac{k}{d}\alpha(\alpha - 1)x_i \right)
$$

$$
\cdot \exp\left( -\frac{1}{2\sigma^2} \cdot (\alpha - 1) \left( d \left( \frac{k\alpha}{d} \right)^2 - 2k\frac{k\alpha}{d} \right) \right).
$$

The middle terms in the numerator and denominator are the same so they cancel out, leaving us with

$$
D_\alpha(Q' \| P) = \frac{1}{\alpha - 1} \log \left( (2\pi\sigma^2)^{-\frac{d}{2}} \int_{\mathbb{R}^d} \frac{\left[ \prod_{i=1}^d \exp\left( -\frac{1}{2\sigma^2} x_i^2 \right) \right]^\alpha}{\left[ \frac{1}{|S_{d,k}|} \sum_{\mu \in S_{d,k}} \prod_{i=1}^d \exp\left( -\frac{1}{2\sigma^2}(x_i - \mu_i)^2 \right) \right]^{\alpha-1}} dx_1 \cdots dx_d \right.
$$

$$
\left. \cdot \frac{\exp\left( -\frac{1}{2\sigma^2} \cdot \alpha d \left( \frac{k}{d}(\alpha - 1) \right)^2 \right)}{\exp\left( -\frac{1}{2\sigma^2} \cdot (\alpha - 1) \left( d \left( \frac{k\alpha}{d} \right)^2 - 2k\frac{k\alpha}{d} \right) \right)} \right)
$$

$$
= \frac{1}{\alpha - 1} \log \left( (2\pi\sigma^2)^{-\frac{d}{2}} \int_{\mathbb{R}^d} \frac{\left[ \prod_{i=1}^d \exp\left( -\frac{1}{2\sigma^2} x_i^2 \right) \right]^\alpha}{\left[ \frac{1}{|S_{d,k}|} \sum_{\mu \in S_{d,k}} \prod_{i=1}^d \exp\left( -\frac{1}{2\sigma^2}(x_i - \mu_i)^2 \right) \right]^{\alpha-1}} dx_1 \cdots dx_d \right)
$$

$$+ \frac{1}{\alpha - 1} \log \left( \frac{\exp\left( -\frac{1}{2\sigma^2} \cdot \alpha d \left( \frac{k}{d}(\alpha - 1) \right)^2 \right)}{\exp\left( -\frac{1}{2\sigma^2} \cdot (\alpha - 1) \left( d \left( \frac{k\alpha}{d} \right)^2 - 2k\frac{k\alpha}{d} \right) \right)} \right)$$

$$= D_\alpha(Q \parallel P) + \frac{1}{\alpha - 1} \log \left( \frac{\exp\left( -\frac{1}{2\sigma^2} \frac{k^2}{d} \alpha(\alpha - 1)^2 \right)}{\exp\left( -\frac{1}{2\sigma^2} \frac{k^2}{d} \alpha(\alpha - 1)(\alpha - 2) \right)} \right)$$

$$= D_\alpha(Q \parallel P) + \frac{1}{\alpha - 1} \log \left( \exp\left( -\frac{1}{2\sigma^2} \frac{k^2}{d} \alpha(\alpha - 1) \right) \right)$$

$$= D_\alpha(Q \parallel P) - \frac{\alpha}{2\sigma^2} \frac{k^2}{d}$$

$$= D_\alpha(Q \parallel P) - \frac{\alpha}{2\sigma^2} \left\| \mathbf{0} - \frac{k}{d} \mathbf{1}_d \right\|_2^2$$

$$= D_\alpha(Q \parallel P) - D_\alpha(Q \parallel Q').$$

$\square$

## A.5. Rényi Divergence between Two Multivariate Gaussians

**Proposition A.1.** *(Gil et al., 2013) Let $f_i = \mathcal{N}(\mu_i, \Sigma)$ and $f_j = \mathcal{N}(\mu_j, \Sigma)$ be two multivariate Gaussian distributions with the same covariance matrix and different means. Then,*

$$D_\alpha(f_i \parallel f_j) = \frac{\alpha}{2} (\mu_i - \mu_j)^\intercal \Sigma^{-1} (\mu_i - \mu_j).$$

**Proposition A.2.** *(Gil et al., 2013) Let $f_i = \mathcal{N}(\mu, \Sigma_i)$ and $f_j = \mathcal{N}(\mu, \Sigma_j)$ be two multivariate Gaussian distributions with the same mean. Then,*

$$D_\alpha(f_i \parallel f_j) = \frac{1}{2(\alpha - 1)} \log \frac{|\Sigma_i|^{1-\alpha} |\Sigma_j|^\alpha}{|(\Sigma_\alpha)^*|}$$

*for*

$$(\Sigma_\alpha)^* = \alpha \Sigma_j + (1 - \alpha) \Sigma_i$$

*whenever $\alpha \Sigma_i^{-1} + (1 - \alpha) \Sigma_j^{-1}$ is positive definite.*

## A.6. Proof of Corollary 3.17

The two distributions are $Q = \mathcal{N}\left( \mathbf{0}, \sigma^2 I \right)$ and $Q' = \mathcal{N}\left( \frac{ck}{d} \mathbf{1}, \sigma^2 I \right)$. Using Proposition A.1,

$$D_\alpha(Q \parallel Q') = \frac{\alpha}{2} \frac{ck}{d} \mathbf{1}^\intercal \left( \sigma^2 I \right)^{-1} \frac{ck}{d} \mathbf{1} = \frac{\alpha c^2 k^2}{2d^2} \sum_{i=1}^d \frac{1}{\sigma^2} = \frac{\alpha c^2 k^2}{2\sigma^2 d}.$$

$\square$

## A.7. Proof of Lemma 3.18

The set $S_{d,k} = \{ v \in \{0,1\}^d \mid \|v\|_1 = k \}$ lies in a subspace of $\mathbb{R}^d$ with dimension $d - 1$ because there is one and only one sum constraint on the entries, which removes one degree of freedom. Since we also offset $S_{d,k}$ by its mean when forming the mixture centers of $P''$, the zero vector also lies in the same subspace as $S_{d,k} - \mu$. The following lemma establishes that we can instead look at Renyi divergence on the $d - 1$-dimensional subspace instead of $\mathbb{R}^d$.

**Lemma A.3.** *Let $S_{d-1}$ be a set of points in $\mathbb{R}^{d-1}$ and $S_d$ be the same set of points in $\mathbb{R}^d$, where each point's last component is $0$. Then,*

$$D_\alpha \left( \mathcal{N}\left( 0, \sigma^2 I_{d-1} \right) \,\middle\|\, \frac{1}{|S_{d-1}|} \sum_{\mu \in S_{d-1}} \mathcal{N}\left( \mu, \sigma^2 I_{d-1} \right) \right) = D_\alpha \left( \mathcal{N}\left( 0, \sigma^2 I_d \right) \,\middle\|\, \frac{1}{|S_{d-1}|} \sum_{\mu \in S_d} \mathcal{N}\left( \mu, \sigma^2 I_d \right) \right)$$

*for all $\alpha \geq 2$.*

*Proof.* For notational convenience, let $x_{1:d-1}$ be the vector formed by taking the first $d-1$ components of $x$, where $x \in \mathbb{R}^d$.

$$
D_\alpha \left( \mathcal{N}\left(0, \sigma^2 I_d\right) \,\Big\|\, \frac{1}{|S_{d-1}|} \sum_{\mu \in S_d} \mathcal{N}\left(\mu, \sigma^2 I_d\right) \right)
$$

$$
= \frac{1}{\alpha-1} \log \int (2\pi\sigma^2)^{\frac{d}{2}} \frac{\exp\left(-\frac{1}{2\sigma^2}\|x\|_2^2\right)^\alpha}{\left(\frac{1}{|S_{d-1}|}\sum_{\mu \in S_d} \exp\left(-\frac{1}{2\sigma^2}\|x-\mu\|_2^2\right)\right)^{\alpha-1}} \, dx_1 \ldots dx_d
$$

$$
= \frac{1}{\alpha-1} \log \int (2\pi\sigma^2)^{\frac{d}{2}} \frac{\left(\exp\left(-\frac{1}{2\sigma^2}\left(\|x_{1:d-1}\|_2^2 + x_d^2\right)\right)\right)^\alpha}{\left(\frac{1}{|S_{d-1}|}\sum_{\mu \in S_d} \exp\left(-\frac{1}{2\sigma^2}\left(\|x_{1:d-1}-\mu_{1:d-1}\|_2^2 + (x_d - \mu_d)^2\right)\right)\right)^{\alpha-1}} \, dx_1 \ldots dx_d
$$

$$
= \frac{1}{\alpha-1} \log \int (2\pi\sigma^2)^{\frac{d}{2}} \frac{\left(\exp\left(-\frac{1}{2\sigma^2}\|x_{1:d-1}\|_2^2\right)\exp\left(-\frac{1}{2\sigma^2}x_d^2\right)\right)^\alpha}{\left(\frac{1}{|S_{d-1}|}\sum_{\mu \in S_d} \exp\left(-\frac{1}{2\sigma^2}\|x_{1:d-1}-\mu_{1:d-1}\|_2^2\right)\exp\left(-\frac{1}{2\sigma^2}x_d^2\right)\right)^{\alpha-1}} \, dx_1 \ldots dx_d
$$

$$
= \frac{1}{\alpha-1} \log \int (2\pi\sigma^2)^{\frac{d}{2}} \frac{\left(\exp\left(-\frac{1}{2\sigma^2}\|x_{1:d-1}\|_2^2\right)\right)^\alpha}{\left(\frac{1}{|S_{d-1}|}\sum_{\nu \in S_{d-1}} \exp\left(-\frac{1}{2\sigma^2}\|x_{1:d-1}-\nu\|_2^2\right)\right)^{\alpha-1}} \exp\left(-\frac{1}{2\sigma^2}x_d^2\right) \, dx_1 \ldots dx_d
$$

$$
= \frac{1}{\alpha-1} \log \int (2\pi\sigma^2)^{\frac{d-1}{2}} \frac{\left(\exp\left(-\frac{1}{2\sigma^2}\|x_{1:d-1}\|_2^2\right)\right)^\alpha}{\left(\frac{1}{|S_{d-1}|}\sum_{\nu \in S_{d-1}} \exp\left(-\frac{1}{2\sigma^2}\|x_{1:d-1}-\nu\|_2^2\right)\right)^{\alpha-1}} \, dx_1 \ldots dx_{d-1}
$$

$$
= D_\alpha \left( \mathcal{N}\left(0, \sigma^2 I_{d-1}\right) \,\Big\|\, \frac{1}{|S_{d-1}|} \sum_{\mu \in S_{d-1}} \mathcal{N}\left(\mu, \sigma^2 I_{d-1}\right) \right).
$$

Below is the proof for Lemma 3.18.

Let $P''' = \mathcal{N}\left(\mathbf{0}, \sigma^2 \exp\left(\frac{c^2 k(d-k)}{\sigma^2 d^2}\right)\right)$. We first establish that $P''(\mathbf{0}) = P'''(\mathbf{0}) \leq Q''(\mathbf{0})$. To see that fact, the set $S_{d,k} - \mu$ is a set that contains all vectors with $k$ entries equal to $\frac{c(d-k)}{d}$ and $d-k$ entries equal to $-\frac{ck}{d}$, so

$$
P''(\mathbf{0}) = (2\pi\sigma^2)^{-\frac{d}{2}} \exp\left(-\frac{1}{2\sigma^2}\left(\frac{c^2(d-k)^2}{d^2}\cdot k + \frac{c^2 k^2}{d^2}\cdot(d-k)\right)\right) = (2\pi\sigma^2)^{-\frac{d}{2}} \exp\left(-\frac{1}{2\sigma^2}\frac{c^2 k(d-k)}{d}\right),
$$

$$
P'''(\mathbf{0}) = \left(2\pi\sigma^2 \exp\left(\frac{c^2 k(d-k)}{\sigma^2 d^2}\right)\right)^{-\frac{d}{2}} = (2\pi\sigma^2)^{-\frac{d}{2}} \exp\left(-\frac{c^2 k(d-k)}{2\sigma^2 d}\right) = P''(\mathbf{0}),
$$

$$
Q''(\mathbf{0}) = (2\pi\sigma^2)^{-\frac{d}{2}} \geq P''(\mathbf{0}).
$$

To see the proof, we first look at the simple case when $d = 2$ and $k = 1$ so $S_{d,k}$ contains two elements. By Lemma A.3, we can reduce the dimension by 1 and look at the scalar case where $P''_{d=2} = \frac{1}{2}(\mathcal{N}(-1, \sigma^2)) + \mathcal{N}(1, \sigma^2))$ and $Q''_{d=2} = \mathcal{N}(0, \sigma^2)$, where we w.l.o.g. scale $c$ such that $P''_{d=2}$ has mixture centers at $-1$ and $1$ since scaling $c$ by $\beta$ is equivalent to scaling $\sigma$ by $\frac{1}{\beta}$. Then, $P'''_{d=2}$ is a Gaussian centered at 0 and having variance $\sigma^2 \exp\left(\frac{1}{\sigma^2}\right) > \sigma^2$ such that $P''_{d=2}(0) = P'''_{d=2}(0)$. Since $P'''_{d=2}$ has smaller variance compared to $P'''_{d=3}$ and has mixture centers that surround 0 on both sides, as $|x| \to \infty$, for small values of $|x|$, $P''_{d=2}(x)$ either increases, or decreases slower than $P'''_{d=2}(x)$, and for large values of $|x|$, $P''_{d=2}(x)$ decreases much faster than $P'''_{d=2}(x)$. Combined with the fact that $P''_{d=2}(0) = P'''_{d=2}(0)$, this means there exists a constant $a$ such that $P''_{d=2}(x) \geq P'''_{d=2}(x)$ for $x \in [0, a]$ and $P''_{d=2}(x) < P'''_{d=2}(x)$ for $x \in (a, \infty)$. To complete the argument,

$$
D_\alpha(Q''_{d=2} \,\|\, P'''_{d=2}) - D_\alpha(Q''_{d=2} \,\|\, P''_{d=2})
$$

$$= \int_{-\infty}^{\infty} \frac{Q''_{d=2}(x)^{\alpha}}{P'''_{d=2}(x)^{\alpha-1}} - \frac{Q''_{d=2}(x)^{\alpha}}{P''_{d=2}(x)^{\alpha-1}} \, dx$$

$$= \int_{-\infty}^{\infty} \int_{P'''_{d=2}(x)}^{P''_{d=2}(x)} \frac{d}{du} \frac{Q''_{d=2}(x)^{\alpha}}{u^{\alpha-1}} \, du dx \qquad \text{by the Fundamental Theorem of Calculus}$$

$$= \int_{-\infty}^{\infty} \int_{P'''_{d=2}(x)}^{P''_{d=2}(x)} (\alpha - 1) \left( \frac{Q''_{d=2}(x)}{u} \right)^{\alpha} \, du dx$$

$$= 2(\alpha - 1) \left( \int_{0}^{a} \int_{P'''_{d=2}(x)}^{P''_{d=2}(x)} \left( \frac{Q''_{d=2}(x)}{u} \right)^{\alpha} \, du dx - \int_{a}^{\infty} \int_{P''_{d=2}(x)}^{P'''_{d=2}(x)} \left( \frac{Q''_{d=2}(x)}{u} \right)^{\alpha} \, du dx \right) = (\star).$$

Since $\frac{Q''_{d=2}(x)}{P''_{d=2}(x)}$ and $\frac{Q''_{d=2}(x)}{P'''_{d=2}(x)}$ are both decreasing functions in $x$ for $x \geq 0$, we have for $x_0 \in [0, a]$ and $x_1 \in (a, \infty)$,

$$\frac{Q''_{d=2}(x_0)}{P'''_{d=2}(x_0)} \geq \frac{Q''_{d=2}(x_0)}{P''_{d=2}(x_0)} \geq \frac{Q''_{d=2}(x_1)}{P''_{d=2}(x_1)} \geq \frac{Q''_{d=2}(x_1)}{P'''_{d=2}(x_1)},$$

so the first integrand is always greater than or equal to the second integrand in $(\star)$. Finally, since $P''_{d=2}$ and $P'''_{d=2}$ are probability densities and need to integrate to the same value of 1,

$$\int_{0}^{a} \int_{P'''_{d=2}(x)}^{P''_{d=2}(x)} \, du dx - \int_{a}^{\infty} \int_{P''_{d=2}(x)}^{P'''_{d=2}(x)} \, du dx = 0,$$

so it follows that $(\star) \geq 0$. It is worthwhile to note the intuition behind the proof. Of $Q''$, $P''$, and $P'''$, the distribution $Q''$ assigns the most probability mass near 0. So, to minimize $D(Q'' \parallel P^*)$ over $P^*$, $P^*$ needs to assigns more probability mass near 0. And between $P''$ and $P'''$, the distribution $P''$ is the one that assigns more mass around 0 because they have the same value at 0, but moving away from 0 in any direction is closer to (at least) one of $P'''$'s mixture centers, so around 0, $P''$ always dominates $P'''$ so $D(Q'' \parallel P'') \leq D(Q'' \parallel P''')$. The same argument holds true for arbitrary $d, k$ since the mixture centers of $P''$ surrounds 0, so along any ray from 0, there exists $a$ such that $P''(x) \geq P'''(x)$ for $x \in [0, a]$, and the claim follows. $\qquad \square$

### A.8. Proof of Corollary 3.19

In the context of Proposition A.2, we have

$$\Sigma_i = \sigma^2 I_d, \qquad \Sigma_j = \sigma^2 \exp\left( \frac{c^2 k(d-k)}{\sigma^2 d^2} \right) I_d, \qquad (\Sigma_\alpha)^* = \sigma^2 \left( \alpha \exp\left( \frac{c^2 k(d-k)}{\sigma^2 d^2} \right) + (1 - \alpha) \right) I_d,$$

$$|(\Sigma_\alpha)^*| = \sigma^{2d} \left( \alpha \exp\left( \frac{c^2 k(d-k)}{\sigma^2 d^2} \right) + (1 - \alpha) \right)^d, \qquad |\Sigma_i|^{1-\alpha} = \sigma^{2d(1-\alpha)}, \qquad |\Sigma_j|^{\alpha} = \sigma^{2d\alpha} \exp\left( \frac{c^2 k(d-k)\alpha}{\sigma^2 d} \right).$$

So, by Lemma 3.18 and Proposition A.2,

$$D_\alpha(Q'' \parallel P'') \leq D_\alpha\left( Q'' \,\middle\|\, \mathcal{N}\left( \mathbf{0}, \sigma^2 \exp\left( \frac{c^2 k(d-k)}{\sigma^2 d^2} \right) I_d \right) \right)$$

$$= \frac{1}{2(\alpha - 1)} \log \frac{|\Sigma_i|^{1-\alpha} |\Sigma_j|^{\alpha}}{|(\Sigma_\alpha)^*|} = \frac{1}{2(\alpha - 1)} \log \frac{\exp\left( \frac{\alpha c^2 k(d-k)}{\sigma^2 d} \right)}{\left( \alpha \exp\left( \frac{c^2 k(d-k)}{\sigma^2 d^2} \right) + (1 - \alpha) \right)^d}.$$

$\qquad \square$

## A.9. Proof of Theorem 3.4

Let $T \in [d]^{|S|}$ be a random vector with $|S|$ entries, where the $k$th entry denotes which submodel is used to train data point $k$. Let $f : \mathcal{S} \times [d]^{|S|} \to \mathbb{R}^m$ be the gradient function that takes in the dataset and the model correspondence for the first $|S|$ data points, and outputs the resulting non-noisy sum of gradients. That means, the output of $f(S,T)$ is deterministic and $f(S',T)$ is a random value that depends on which model is assigned to the last data point $x$. The function $f$ satisfies bounded difference as a result of gradient norm clipping of $c$. That is,

$$\|f(S',T) - f(S,T)\|_2 \le c$$

for all adjacent datasets $S, S'$ and $T$. We can express the output of $\mathcal{M}$ as a mixture of distributions,

$$\mathcal{M}(S) = \sum_T p_T \mathcal{N}\left(f(S,T), \sigma^2 I\right),$$

$$\mathcal{M}(S') = \sum_T p_T \mathcal{N}\left(f(S',T), \sigma^2 I\right).$$

By the quasi-convexity of Renyi divergence,

$$D_\alpha(\mathcal{M}(S) \,\|\, \mathcal{M}(S')) \le \sup_T D_\alpha\left(\mathcal{N}\left(f(S,T), \sigma^2 I\right) \,\|\, \mathcal{N}\left(f(S',T), \sigma^2 I\right)\right)$$

$$= \sup_T D_\alpha\left(\mathcal{N}\left(\mathbf{0}, \sigma^2 I\right) \,\|\, \mathcal{N}\left(f(S',T) - f(S,T), \sigma^2 I\right)\right).$$

where the last line comes as a result of the translational invariance of Renyi divergence.

The first term is a Gaussian centered at $\mathbf{0}$ and the second term is a mixture of Gaussians centered at the possible outcomes of $f(S',T) - f(S,T)$, which has $d$ different outcomes corresponding to which of the $d$ models the data point $x$ is trained on, each with probability $\frac{1}{d}$. The $d$ different outcomes have disjoint support because the model parameters of the $d$ submodels are disjoint. Let the non-zero parts of these $d$ outcomes be $c_1, \ldots, c_d$ such that $\|c_k\| \le c$ for all $1 \le k \le d$. Since the covariances are symmetric, we can apply a rotation to each of the $d$ different supports—and leave the first term $\mathcal{N}\left(\mathbf{0}, \sigma^2 I\right)$ unchanged—so that $c_k$ becomes $\|c_k\| \mathbf{e}_1$, where $\mathbf{e}_1$ is the first standard basis. Now in each of the $d$ supports, the first and second terms are both product distributions that are different in the first entry and identical in the rest, so we can drop the coordinates where they are the same. The divergence thus becomes

$$D_\alpha(\mathcal{M}(S) \,\|\, \mathcal{M}(S')) \le \sup_{\|c_1\| \le c, \ldots, \|c_d\| \le c} D_\alpha\left(\mathcal{N}(\mathbf{0}, \sigma^2 I) \,\Big\|\, \frac{1}{d}\sum_{i=1}^d \mathcal{N}\left(\|c_i\|\mathbf{e}_i, \sigma^2 I\right)\right).$$

Setting $\|c_1\| = \cdots \|c_d\| = c$ would achieve the supremum since they give the maximum divergence from $\mathbf{0}$, so

$$D_\alpha(\mathcal{M}(S) \,\|\, \mathcal{M}(S')) \le D_\alpha\left(\mathcal{N}(\mathbf{0}, \sigma^2 I) \,\Big\|\, \frac{1}{d}\sum_{i=1}^d \mathcal{N}\left(c\,\mathbf{e}_i, \sigma^2 I\right)\right)$$

where the first term matches $Q$ and the second term matches $P$ in Theorem 3.1 where $k = 1$. Switching the order of $\mathcal{M}(S)$ and $\mathcal{M}(S')$ requires a similar analysis, with the first and second terms switched. Thus,

$$\epsilon = \max\left\{D_\alpha(\mathcal{M}(S) \,\|\, \mathcal{M}(S')), D_\alpha(\mathcal{M}(S') \,\|\, \mathcal{M}(S))\right\}$$

$$\le \max\left\{D_\alpha(P \,\|\, Q), D_\alpha(Q \,\|\, P)\right\}.$$

$\square$

## A.10. Proof of Theorem 3.5

If we condition on how the model is split into submodels, then the privacy guarantee is given by Theorem 3.4, which does not depend on how the model is split. Therefore, if we do not condition on how the model is split, the resulting distribution of the submodel is a mixture of the conditional distributions, so the resulting distribution of the gradients is a mixture of the resulting conditional distributions of gradients, conditioned on how the model is split. Thus, by the quasi-convexity of Rényi divergence, the divergence between the unconditional distributions of gradients (resulting from using $S$ vs $S'$ for the dataset) is upper-bounded by the max divergence between the conditional distributions of gradients, which is given in Theorem 3.4.

$\square$

## A.11. Proof of Theorem 3.7

The full proof has a lot of similarities to the proof of Theorem 3.4, which we will not repeat but only focus on the different part.

We first consider the case of one dropout layer then the case of multiple dropout layers.

For one dropout layer, if we let the nodes of the dropout layer be $v_1, \ldots, v_n$ and let the set of weights whose gradients are set to 0 by dropping node $v_k$ be $W_k$ for $k = 1, \ldots, n$, then $W_1, \ldots, W_n$ are disjoint sets that cover the parameters of the preceding and succeeding layers. Let the combined parameters of the preceding and succeeding layers be $W$ such that $W = \bigcup_{k=1}^n W_k$. The set of weights $W$ can be treated as a full model for the scope of this proof since it has its own gradient clipping norm of $c$. Consider a particular outcome of the dropout layer that keeps nodes $\{v_j\}_{j \in J}$ for some $J \subseteq [n]$, which is equivalent to forming a submodel $W_J = \bigcup_{j \in J} W_j$. Consider the complementary submodel $W_J^c = \bigcup_{j \notin J} W_j = W \setminus W_J$. Then, conditioned on the event that the submodel formed by the dropout layer is either $W_J$ or $W_J^c$, each has a conditional probability of 0.5 because each has the same unconditional probability of $(0.5)^n$, so $W_J$ and $W_J^c$ form one way to partition the model. Then, the dropout layer is a mixture of $W_J$ and $W_J^c$ for all possible $J$, and thus Theorem 3.5 applies where $d = 2$.

For the case of multiple dropout layers, if the dropout layers have no overlapping parameters whose gradients are directly affected (i.e., whose gradient will be 0 if a node it connects to is dropped out), then the analysis remains the same. If there are, for example in the case where two dropout layers are applied to the input and output of another layer, then conditioned on $J$ (where $J$ is now a subset of nodes in the union of the two dropout layers), the submodels $\bigcup_{j \in J} W_j$ and $\bigcup_{j \notin J} W_j$ no longer cover $W$ but are still disjoint, so the analysis to quantify the divergence between the two conditional distributions of gradients (conditioned on $J$) using Theorem 3.1 is still valid, since the gradients not covered by either the submodels are 0 regardless of which dataset is used, so we can throw away these coordinates in analyzing the Rényi divergence. $\qquad\square$

## A.12. Proof of Theorem 3.8

Consider two adjacent datasets $S, S' \in \mathcal{S}$, where $S' = S \cup \{x\}$. Let $A \in \mathbb{R}^{|S'| \times T}$ be the random sampling matrix, whose entries are binary and each row contains $k$ 1s and $T - k$ 0s, where the entry at $i, t$ is 1 if the $i$th data point is used in the $t$th iteration, and 0 otherwise. Let $A_S$ be the first $|S|$ rows of $A$ and $A_{|S'|}$ be the last row of $A$, where $A_{|S'|}$ will be disregarded if the dataset is $S$. Let $\mathbb{P}$ denote the implied distribution of any function of $A$. Let $d$ be the number of parameters in the neural network with a gradient. Let $Y_t \in \mathbb{R}^d$ be the noisy sum of gradients in iteration $t$. Let $g_t$ be the gradient function of the $t$th iteration, which take in as inputs $S$ or $S'$ (the dataset), $A_{\cdot, t}$ (the sampling mask used in this iteration), and $y_1, \ldots, y_{t-1}$ (the previous gradients, or equivalently, the previous model updates), and satisfies that

$$\|g_t(S, A_{\cdot, t}; y_1, \ldots, y_t) - g_t(S', A_{\cdot, t}; y_1, \ldots, y_t)\|_2 \le c$$

for all $t, A, y_1, \ldots, y_t$, which follows from $l_2$ norm clipping of each gradient. Let $P_0(Y_1, \ldots, Y_T)$ be the joint distribution of the noise gradients if $S$ is used, and $Q_0(Y_1, \ldots, Y_T)$ if $S'$ is used. Their distributions can be written as

$$P_0(Y_1, \ldots, Y_T) = \sum_{A_S} \mathbb{P}(A_S) P_0(Y_1, \ldots, Y_T \mid A_S)$$

and

$$Q_0(Y_1, \ldots, Y_T) = \sum_{A_S} \mathbb{P}(A_S) \sum_{A_{|S'|}} \mathbb{P}\left(A_{|S'|}\right) Q_0(Y_1, \ldots, Y_T \mid A_S, A_{|S'|}).$$

Since Renyi divergence is quasi-convex, the Renyi divergence between $P_0(Y_1, \ldots, Y_T)$ and $Q_0(Y_1, \ldots, Y_T)$ is upper bounded by

$$D_\alpha(P_0 \| Q_0) \le \sup_{A_S} D_\alpha\left(P_0(Y_1, \ldots, Y_T \mid A_S) \,\Big\|\, \sum_{A_{|S'|}} \mathbb{P}\left(A_{|S'|}\right) Q_0(Y_1, \ldots, Y_T \mid A)\right)$$

$$= \sup_{A_S} \frac{1}{\alpha - 1} \log \int \frac{P_0(y_1, \ldots, y_T \mid A_S)^\alpha}{\left(\sum_{A_{|S'|}} \mathbb{P}\left(A_{|S'|}\right) Q_0(y_1, \ldots, y_T \mid A)\right)^{\alpha-1}} \, dy_1 \ldots dy_T$$

$$= \sup_{A_S} \frac{1}{\alpha - 1} \log \int \frac{\prod_{t=1}^{T} P_0(y_t \mid y_1, \ldots, y_{t-1}, A_S)^{\alpha}}{\left( \sum_{A_{|S'|}} \mathbb{P}\left(A_{|S'|}\right) \prod_{t=1}^{T} Q_0(y_t \mid y_1, \ldots, y_{t-1}, A)\right)^{\alpha - 1}} \, dy_1 \ldots dy_T = (\star).$$

The conditional distributions in the numerator and denominator are

$$P_0(y_t \mid y_1, \ldots, y_{t-1}, A_S) = \mathcal{N}(g_t(S, A_{\cdot,t}; y_1, \ldots, y_{t-1}), \ \sigma^2 I_d),$$

$$Q_0(y_t \mid y_1, \ldots, y_{t-1}, A) = \mathcal{N}(g_t(S', A_{\cdot,t}; y_1, \ldots, y_{t-1}), \ \sigma^2 I_d).$$

Consider the substitution

$$x_t = y_t - g_t(S, A_{\cdot,t}; y_1, \ldots, y_{t-1})$$

so that

$$P_0(x_t \mid y_1, \ldots, y_{t-1}, A_S) = \mathcal{N}(\mathbf{0}, \ \sigma^2 I_d),$$

$$Q_0(x_t \mid y_1, \ldots, y_{t-1}, A) = \mathcal{N}(g_t(S', A_{\cdot,t}; y_1, \ldots, y_{t-1}) - g_t(S, A_{\cdot,t}; y_1, \ldots, y_{t-1}), \ \sigma^2 I_d)$$

$$= \begin{cases} \mathcal{N}(c_t v_t, \ \sigma^2 I_d) & A_{|S'|,t} = 1 \\ \mathcal{N}(\mathbf{0}, \ \sigma^2 I_d) & A_{|S'|,t} = 0 \end{cases}$$

$$= \mathcal{N}\left(c_t v_t \mathbb{1}_{\left\{A_{|S'|,t}=1\right\}}, \ \sigma^2 I_d\right)$$

where $c_t \leq c$ and $\|v_t\|_2 = 1$, both of which may depend on $A, y_1, \ldots, y_{t-1}$. Also, since there is a bijection between $x_t$ and $y_t$ given $A, y_1, \ldots, y_{t-1}$, by a recursive argument, there is a bijection between $x_1, \ldots, x_t$ and $y_1, \ldots, y_t$ given $A$ for all $t$. Thus, the conditioning on $y_1, \ldots, y_{t-1}$ is equivalent to conditioning on $x_1, \ldots, x_{t-1}$, so we can write

$$P_0(y_t \mid y_1, \ldots, y_{t-1}, A_S) = P_0(y_t \mid x_1, \ldots, x_{t-1}, A_S),$$

$$Q_0(y_t \mid y_1, \ldots, y_{t-1}, A) = Q_0(y_t \mid x_1, \ldots, x_{t-1}, A).$$

Lastly, for each variable $y_t$ in the integration in $(\star)$, we replace it by $y_t - g_t(S, A_{\cdot,t}; y_1, \ldots, y_{t-1})$. The integration over $y_t$ is on $\mathbb{R}^d$ so the new integration region over $y_t - g_t(S, A_{\cdot,t}; y_1, \ldots, y_{t-1})$ is still on $\mathbb{R}^d$. Then we apply the substitutions,

$$(\star) = \sup_{A_S} \frac{1}{\alpha - 1} \log \int \frac{\prod_{t=1}^{T} P_0(x_t \mid x_1, \ldots, x_{t-1}, A_S)^{\alpha}}{\left( \sum_{A_{|S'|}} \mathbb{P}\left(A_{|S'|}\right) \prod_{t=1}^{T} Q_0(x_t \mid x_1, \ldots, x_{t-1}, A)\right)^{\alpha - 1}} \, dx_1 \ldots dx_T$$

$$= \sup_{A_S} \frac{1}{\alpha - 1} \log \int \frac{\prod_{t=1}^{T} \mathcal{N}(\mathbf{0}, \sigma^2 I_d)^{\alpha}}{\left( \sum_{A_{|S'|}} \mathbb{P}\left(A_{|S'|}\right) \prod_{t=1}^{T} \mathcal{N}(c_t v_t \mathbb{1}_{\left\{A_{|S'|,t}=1\right\}}, \ \sigma^2 I_d)\right)^{\alpha - 1}} \, dx_1 \ldots dx_T$$

$$= \sup_{A_S} D_{\alpha}\left( \prod_{t=1}^{T} \mathcal{N}(\mathbf{0}, \sigma^2 I_d) \,\Bigg\|\, \sum_{A_{|S'|}} \mathbb{P}\left(A_{|S'|}\right) \prod_{t=1}^{T} \mathcal{N}(c_t v_t \mathbb{1}_{\left\{A_{|S'|,t}=1\right\}}, \ \sigma^2 I_d) \right)$$

$$\leq \sup_{c_t \leq c, \|v_t\|_2 \leq 1} D_{\alpha}\left( \prod_{t=1}^{T} \mathcal{N}(\mathbf{0}, \sigma^2 I_d) \,\Bigg\|\, \sum_{A_{|S'|}} \mathbb{P}\left(A_{|S'|}\right) \prod_{t=1}^{T} \mathcal{N}(c_t v_t \mathbb{1}_{\left\{A_{|S'|,t}=1\right\}}, \ \sigma^2 I_d) \right) = (\star\star)$$

where in the second-to-last line, $\mathbb{P}\left(A_{|S'|}\right)$ is a constant that does not depend on the choice of $A_S$, and the only item that depends on $A_S$ is $c_t v_t$. The supremum is achieved by the choice of $c_t$ and $v_t$ such that $c_t = c$ and $v_t = \mathbf{e}_1$ (or any vector with norm 1, but all are equivalent due to the rotation symmetry of $\prod_{t=1}^{T} \mathcal{N}(\mathbf{0}, \sigma^2 I_d)$ around 0). Then, since $x_1, \ldots, x_T$ are either $\mathbf{e}_1$ or $\mathbf{0}$ under $Q_0$ and $\mathbf{0}$ under $P_0$, we only need to integrate over their first components and ignore the rest $d - 1$ components since the rest have the same distribution under $P_0$ and $Q_0$. So,

$$(\star\star) \leq D_{\alpha}\left( \prod_{t=1}^{T} \mathcal{N}(0, \sigma^2) \,\Bigg\|\, \sum_{A_{|S'|}} \mathbb{P}\left(A_{|S'|}\right) \prod_{t=1}^{T} \mathcal{N}(c \mathbb{1}_{\left\{A_{|S'|,t}=1\right\}}, \ \sigma^2) \right)$$

$$= D_\alpha \left( \prod_{t=1}^{T} \mathcal{N}(0, \sigma^2) \,\Big\|\, \sum_{\mu \in S_{T,k}} \frac{1}{|S_{t,k}|} \prod_{t=1}^{T} \mathcal{N}(c\mathbb{1}\{\mu_t = 1\}, \sigma^2) \right)$$

$$= D_\alpha \left( \mathcal{N}(\mathbf{0}, \sigma^2 I_T) \,\Big\|\, \sum_{\mu \in S_{T,k}} \frac{1}{|S_{t,k}|} \mathcal{N}(c\mu, \sigma^2 I_T) \right)$$

$$= D_\alpha(P \,\|\, Q).$$

The same proof goes for $D_\alpha(Q \,\|\, P)$ by switching the numerator and denominator, and taking the maximum of the two completes the proof.

# B. More Graphs.

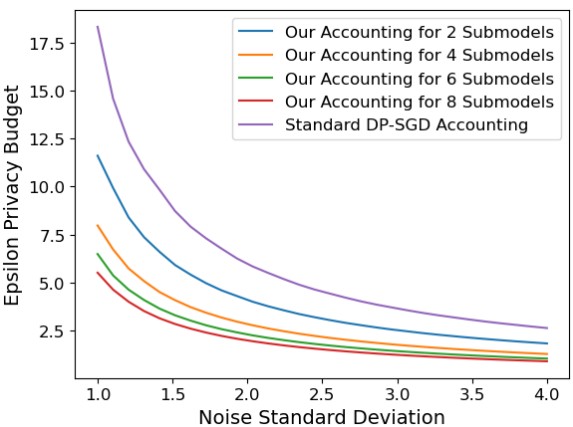

(a) The graph compares the noise standard deviation vs the $\epsilon$ privacy cost in $(\epsilon, \delta)$-DP with and without applying our amplification, where $\delta = 10^{-5}$ and we train for 400 iterations. The data is Poisson subsampled with rate 0.1. Lower $\epsilon$ and less noise are better.

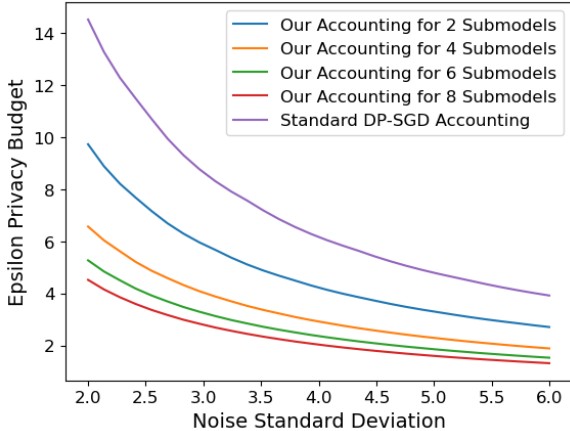

(b) The graph compares the noise standard deviation vs the $\epsilon$ privacy cost in $(\epsilon, \delta)$-DP with and without applying our amplification, where $\delta = 10^{-5}$ and we train for 2000 iterations. The data is Poisson subsampled with rate 0.1. Lower $\epsilon$ and less noise are better.

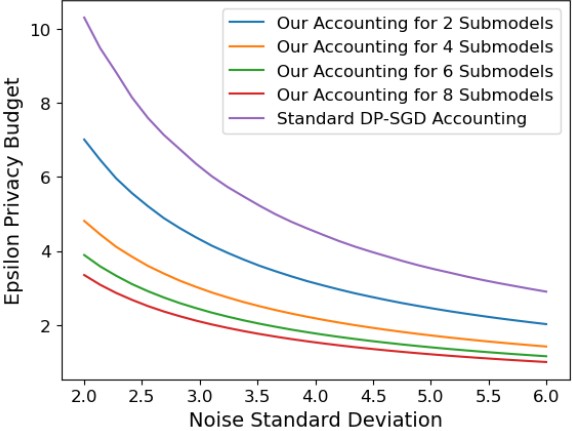

(c) The graph compares the noise standard deviation vs the $\epsilon$ privacy cost in $(\epsilon, \delta)$-DP with and without applying our amplification, where $\delta = 10^{-6}$ and we train for 1500 iterations. The data is Poisson subsampled with rate 0.08. Lower $\epsilon$ and less noise are better.

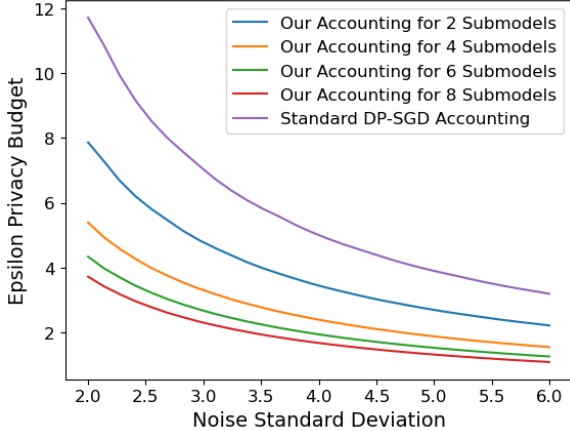

(d) The graph compares the noise standard deviation vs the $\epsilon$ privacy cost in $(\epsilon, \delta)$-DP with and without applying our amplification, where $\delta = 10^{-5}$ and we train for 600 iterations. The data is Poisson subsampled with rate 0.15. Lower $\epsilon$ and less noise are better.

*Figure 5.* More comparison for disjoint submodel splitting.

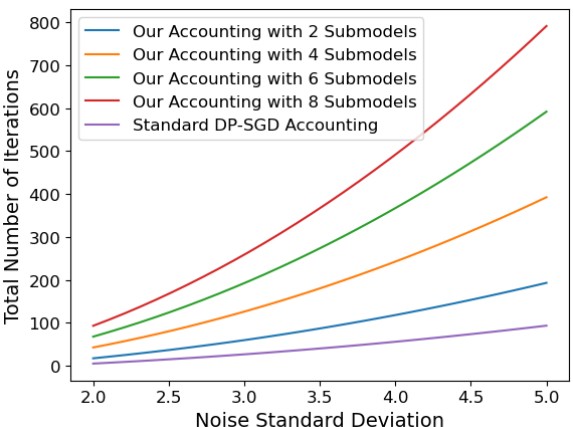

(a) The graph compares the noise standard deviation vs the number of training iterations with and without applying our amplification, where $(\epsilon, \delta) = (1, 10^{-5})$. The data is Poisson subsampled with rate $0.1$. More iterations and less noise are better.

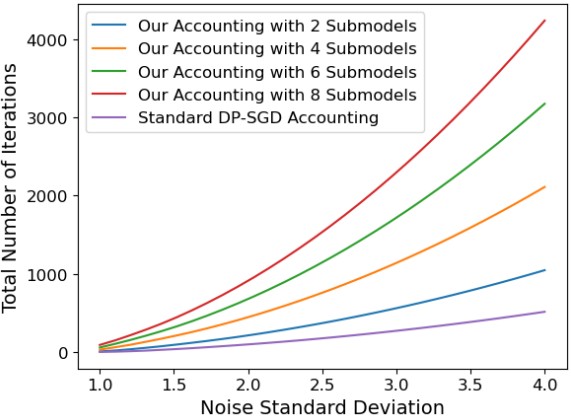

(b) The graph compares the noise standard deviation vs the number of training iterations with and without applying our amplification, where $(\epsilon, \delta) = (3, 10^{-5})$. The data is Poisson subsampled with rate $0.1$. More iterations and less noise are better.

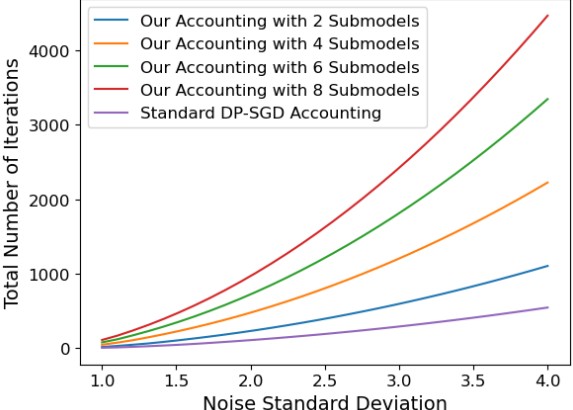

(c) The graph compares the noise standard deviation vs the number of training iterations with and without applying our amplification, where $(\epsilon, \delta) = (5, 10^{-6})$. The data is Poisson subsampled with rate $0.15$. More iterations and less noise are better.

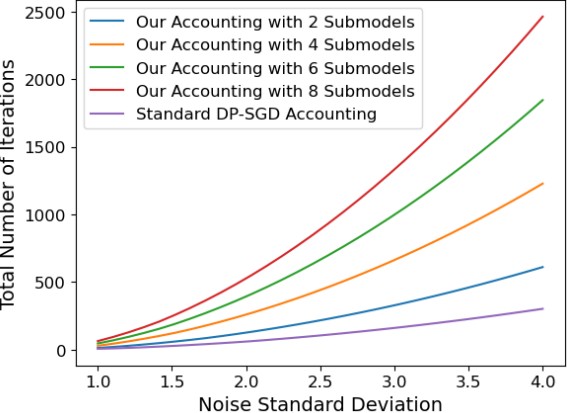

(d) The graph compares the noise standard deviation vs the number of training iterations with and without applying our amplification, where $(\epsilon, \delta) = (8, 10^{-6})$. The data is Poisson subsampled with rate $0.3$. More iterations and less noise are better.

*Figure 6.* More comparison for disjoint submodel splitting.

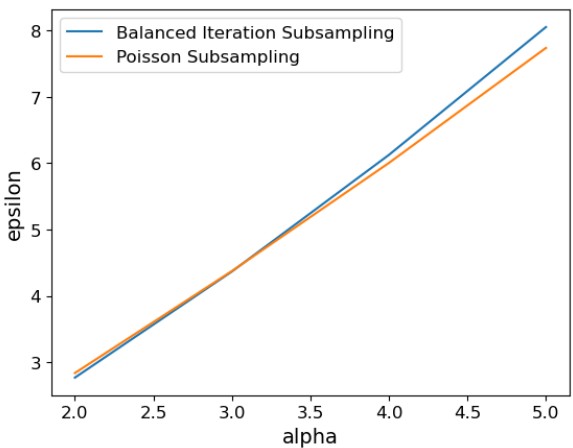

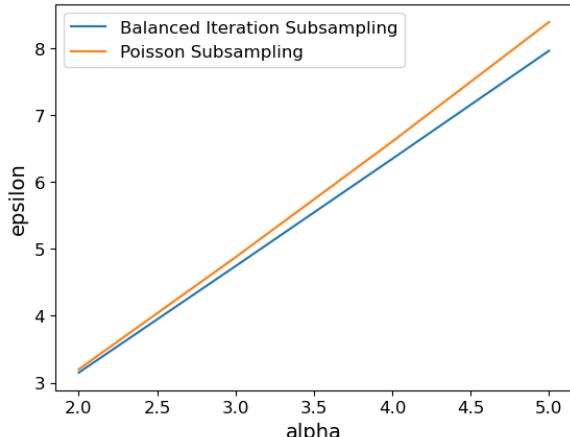

(a) The $\epsilon$ vs $\alpha$ graph for Balanced Iteration Subsampling and Poisson Subsampling, where $T = 1000$, $k = 100$, and $\sigma = 2$. Lower $\epsilon$ is better.

(b) The $\epsilon$ vs $\alpha$ graph for Balanced Iteration Subsampling and Poisson Subsampling, where $T = 200$, $k = 100$, and $\sigma = 4$. Lower $\epsilon$ is better.

*Figure 7.* More comparison for Balanced Iteration Subsampling vs Poisson Subsampling.

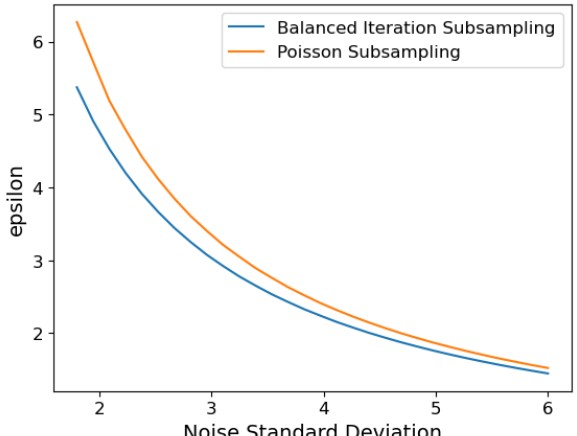

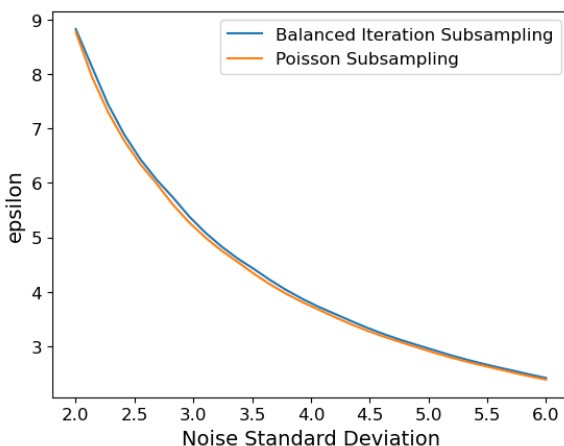

(a) The $\epsilon$ privacy cost (in $(\epsilon, \delta)$-DP) vs noise standard deviation $\sigma$ graph for Balanced Iteration Subsampling and Poisson Subsampling, where $T = 10$, $k = 5$, and $\delta = 10^{-6}$. Lower $\epsilon$ and less noise are better. For small $T$, Balanced Iteration Subsampling can achieve the same privacy guarantee with less noise injection.

(b) The $\epsilon$ privacy cost (in $(\epsilon, \delta)$-DP) vs noise standard deviation $\sigma$ graph for Balanced Iteration Subsampling and Poisson Subsampling, where $T = 1500$, $k = 120$, and $\delta = 10^{-4}$. Lower $\epsilon$ and less noise are better. For large $T$, the two curves are nearly identical.

*Figure 8.* More comparison for Balanced Iteration Subsampling vs Poisson Subsampling.

