# OpenReview forum: "Leveraging Randomness in Model and Data Partitioning for Privacy Amplification"
_ICML.cc/2025/Conference — ICML 2025 poster_

### Official Review · Reviewer_j224 · 2025-03-07

**Overall Recommendation:** 4

**Summary:**

The submission analyzes privacy amplification for Renyi DP in two settings where $N$ (sets of) records are independently assigned to $k$ out of $d$ components of DP-SGD:
(1) Data partitioning, where each record from a dataset of size $N$ contributes to $k$ out of $d$ gradient steps ("balanced iteration subsampling"),
and (2) model partitioning, where all records associated with one of $N$ clients are used to update $k=1$ out of $d$ subsets of model parameters.

The authors first present a general upper bound on the Renyi divergence on the Gaussian mixture distributions that arise from such random partitioning schemes.
A proof for this main theorem is provided at the end of the methods section. The main objective of this proof is to determine sound upper bounds
that eliminate computationally intractable multinomial terms (forward divergence) or eliminate mixture densitities from the denominator in the Renyi divergence integral (reverse divergence).

For model partitioning, the general bound is first instantiated for disjoint partitionings of the model parameters. It is then generalized for non-disjoint partitionings by composing privacy guarantees for the intersection and symmetric difference of the parameter subsets. Next, the disjoint partitioning bound is generalized to probabilistic via joint quasi-convexity of Renyi divergences. Finally, this probabilistic disjoint partitioning bound is applied to the special case of parameter dropout, which partitions models into activated and deactivated neurons.
For data partitioning, the general bound is again instantiated to obtain epoch-level privacy guarantees, which are then compared to a composition of iteration-level privacy guarantees for Poisson subsampling.

In the experimental section, the authors evaluate the utility of ResNet-101 finetuned on CIFAR-10 with (1) model partitioning and bounds from prior work (2) model partitioning with the proposed bounds and (3) without model partitioning. The results show an improved privacy--utility trade-off for (2) compared to (1), achieving an accuracy closer to (3).

**Claims And Evidence:**

The submission makes two central claims:
1. That the randomness inherent to different commonly used model partitioning strategies can be used to provide stronger privacy guarantees.
2. That balanced iteration subsampling can offer stronger privacy amplification than Poisson subsampling for certain parameterizations.

Claim 1 is proven by formally deriving corresponding amplification guarantees.

Claim 2 is proven by deriving corresponding amplification guarantees and comparing them to tight amplification bounds for Poisson subsampling for different epoch lengths, subsampling rates, and fixed RDP parameter $\alpha$ (See Fig. 2).

**Rating: Good**

**Essential References Not Discussed:**

Balanced iteration subsampling with $k=1$ has already been proposed in "Balls-and-Bins Sampling for DP-SGD" by Chua et al. (December 2024).

However, this prior work appeared just one month before the submission deadline, so I do not think that the authors necessarily need to discuss it.

**Experimental Designs Or Analyses:**

As stated above, the evaluation in Section 4 is sound.
However, the privacy utility trade-off is only investigated for a specific model partitioning scheme and one specific choice of privacy budget, and one random seed. The argument for leveraging the randomness in model partitioning in practice could be strengthtened by also:
* Considering a wider range of privacy budgets to show some form of Pareto-domination of the baseline
* Also considering fully disjoint model partitioning and random dropout partitioning.
* Repeating the experiment with multiple random seeds and reporting standard deviations.

Similarly, the evaluation of the RDP profiles in Fig. 3 and Fig. 4 could be repeated for different choices of $T, k, \sigma$ in the appendix to show that the derived bounds are beneficial beyond this very specific  choice of parameters.

Finally, the privacy--utility trade-off attained by balanced iteration subsampling is not investigated. Thus, it is not clear whether there is any benefit to actually using this new data partitioning scheme for model training in practice.

**Rating: Ok for a theory-focused work, but experimental evaluation could be expanded / be more thorough to show impact on practical applications**

**Methods And Evaluation Criteria:**

### Methods
In the case of model partitioning, the work does not provide new methods as such, but rather derives better privacy guarantees for these methods.

The proposed balanced iteration subsampling appears like a reasonable alternative to random shuffling or Poisson sampling. It fulfills its role in demonstrating that amplification-by-subsampling can also be attained by sampling non-i.i.d. lots/batches.
However, it is not clear whether this subsampling method is preferable for model training (see "Experimental Designs or Analyses" below).

### Evaluation Criteria
Privacy guarantees are compared by plotting (Renyi) privacy profiles, which is common practice.
The privacy--utility trade-off attained via the amplification bounds is evaluated by comparing accuracy at a fixed privacy budget, which also makes sense. CIFAR and ResNet are standard choices for such experiments in DP literature.

**Rating: Good**

**Other Comments Or Suggestions:**

### Comments
Theorem 3.4 states that the amplification guarantee holds for any disjoint model splitting method. However, the proof only considers deterministic splitting. Probabilistic splitting is only analyzed later in Theorem 3.5. You may want to add a quantifier: "Each iteration of Algorithm 1 with a >deterministic< disjoint model splitting method [...]".

### Conclusion
Overall, the submission makes substantial, well-presented contributions in two different directions: Privacy accounting for non-i.i.d. subsampling and leveraging internal randomness due to model partitioning.

Since model partitioning is already used for computational reasons, the latter essentially provides additional privacy for free. As such, the work could potentially have high impact on federated learning literature. The main Theorem appears very general and may thus also enable the analysis of various other non-i.i.d. subsampling schemes. Thus, there may also be some impact on differentially private ML in the centralized setting.

My main concern is that the experimental evaluation is not sufficiently thorough to make any definite statements about how large the benefit of using these new sources of randomness actually is in practical applications. However, since the main contribution lies in the conducted theoretical analysis, I nevertheless recommend acceptance.

## Update after Rebuttal
The authors have addressed most of my comments. I continue to recommend acceptance, see details in rebuttal response below.

**Other Strengths And Weaknesses:**

### Other Strengths
* The paper is overall well-written and structured. Especially the main section which first states the main result, then applies it to different use cases, and finally discusses the technical details of its derivation.
* The work clearly states a research question (paragraph 2 of Section 1), which is good.
* The authors are transparent about limitations / avenues towards deriving tighter bounds (e.g., by leveraging the randomness of probabilistic model partitioning in Theorem 3.5)

### Other Weaknesses
* The work uses RDP, which overestimates privacy leakage when converted to approximate differential privacy
* The caption of Fig. 2 states "With different values of $\alpha$, the graph shows a similar pattern." This should be substantiated by actually plotting the graph for different values of $\alpha$ in the appendix.
* The authors do not specify for which neighboring relation they derive their bounds. I assume it's insertion/removal of a single record (of a single client), but it would be good to clarify this.
* In Fig. 1 "No amplification" and Section 4 "baseline", it is not clear how exactly the baseline privacy guarantes are computed. It would be better to clarify this in the appendix.

**Questions For Authors:**

* At the bottom of Fig. 2, there is a very thin green line. Does this mean that balanced iteration subsampling is better for $\gamma \to 0$, or is this a bug in the plotting code?
* Could you please explain why Corollary 3.7 follows from Theorem 3.5, even though the model splitting is now done per client? (See "Theoretical Claims" above).

**Relation To Broader Scientific Literature:**

On a high-level, this work studies privacy amplification, i.e., leveraging elements of internal randomness that induce mixture distributions.

The proposed balanced iteration represents an alternative to commonly used subsampling schemes that sample each batch for a training epoch independently. Similar non-i.i.d. subsampling schemes have already been studied in prior work (shuffling, random check-ins, and balls-and-bins sampling). However, it generalizes them by allowing each record to contribute to multiple training steps. I.e., this aspect is somewhat novel.

It seems like analyzing amplification by model partitioning is completely novel, but I am not sufficiently familiar with this side of the literature to make a definite statement about its novelty.

The privacy analysis is conducted in the framework of Renyi differential privacy / moments accounting. This enables very simple privacy accounting for subsampled mechanisms, but overestimates privacy when converted to approximate DP. However, this framework has been superseded by other numerical and analytical (e.g., Fast Fourier Accounting (Koskela et al, 2019) and Analytical Fourier Accounting (Zhu et al., 2022)) approaches.

**Theoretical Claims:**

I went through the proofs in Section 3.4 and read the proofs of Theorems 3.4, 3.5, 3.7, and 3.8 in detail.
I skimmed the other proofs in the appendix. **Overall, the proof strategy appears sound, but I did not not check each individual equation for correctness**.

I have a minor concern w.r.t. Corollary 3.7 (Parameter dropout). The proof seems to largely follow that of Theorem 3.5 (Disjoint partitioning). However, Theorem 3.5 assumes that the partitioning is done centrally (see l. 172 in Algorithm 1), whereas dropout is applied independently per client (see l. 228 in Algorithm 2).
I believe that the guarantee still holds due to parallel composition (i.e., we only need to apply Theorem 3.5 to the specific client that holds the inserted/removed record), but it would be good if the authors could clarify this.

**Rating: Ok, but requires some clarification**

---

> ### Author Rebuttal · Authors · 2025-04-01
>
> We thank the reviewer for their detailed review of our paper and their valuable feedback.
>
> Our paper is indeed the first to point out and quantify the privacy gain from model parallelism techniques already employed in federated learning.
>
> Following the reviewer's suggestions, we have done more experiments for both centralized and federated settings. In the centralized setting with sample-level $(8, 10^{-5})$-DP,  training ResNet101 with 3 submodels under the standard DP-SGD analysis achieves 79.80\% accuracy and using our analysis accuracy increases to 82.43\% (this is because our analysis accounts for the amplification gain for model splitting, allowing us to add less noise while still achieving the same DP guarantee); 8 submodels under the standard DP-SGD analysis has 76.80\% accuracy and using our analysis accuracy increases to 80.52\%. In federated setting with user-level $(8, 10^{-5})$-DP,  3 submodels with the standard analysis has 78.47\% accuracy and using our analysis accuracy increases to 80.28\%; 8 submodels with the standard analysis has 76.96\% accuracy, and using our analysis accuracy increases to 79.11\%. The experiments are run with 3 random seeds and all have standard deviations of around 0.7\%.
>
> The design choice to not partition the first and last layers of ResNet-101 is guided by insights from the model splitting literature (e.g. Dun et. al.). They argue that since these layers pick up important features, partitioning them leads to degraded accuracy. Following the guidance from this literature, we experimented with partial and not fully disjoint model splitting.
>
> As for the comparison between Balanced Iteration Subsampling and Poisson Subsampling, we would like to emphasize a point which we will make more clear in the revision. Balanced Iteration and Poisson Subsampling achieve similar privacy-accuracy trade-offs experimentally. This is because standard training includes a large number of iterations, and for large numbers of iterations, both the training dynamic and privacy guarantees of the two become comparable as pointed out in the paper. The main advantage of Balanced Iteration Subsampling is that it is more practical and deployment-friendly especially in the federated setting. Poisson Subsampling has implementation-related drawbacks as it gives a variable load to each client, which can overwhelm the resources of the client and undermine fair use policies. The DP community has been interested in studying other (more deployment-friendly) data subsampling techniques whose utility and privacy guarantees are comparable to Poisson Subsampling, for example random shuffling and random check-ins (Balle et al 2020). Our paper fills this gap by showing that this is indeed the case for Balanced Iteration Subsampling. As for the actual experiments, for CIFAR-10 with WideResNet-40-4,  $(8, 10^{-5})$-DP,  2000 iterations, using each sample 655 times for Balanced Iteration Subsampling and with probability $\frac{655}{2000}$ in each iteration for Poisson Subsampling, Balanced Iteration Subsampling injects noise with $\sigma = 10.17$ and achieves validation accuracy of $70.21\%$ (with standard deviation $0.69\%$), while Poisson Subsampling injects noise with $\sigma = 10.20$ and achieves validation accuracy of $70.13\%$ (with standard deviation $0.78\%$). ResNet-101 gives similar results. We will include these experiments and show more graphs to compare the privacy guarantees of the two subsampling methods in the revision.
>
> In Figure 2, the thin line at $\gamma \approx 0$ is due to discretization of $\gamma$ values and the code arbitrarily ruling in favor of Balanced Iteration Subsampling when $\gamma = 0$.
>
> On the concern about Corollary 3.7, we would like to clarify that the privacy gain we utilize comes from the random (and independent) assignment of submodels to the clients. So, although in Algorithm 1, model splitting in done centrally, each client independently receives a submodel. Algorithm 2's parameter dropout can be seen as forming $2^m$ submodels (where $m$ is the number of parameters subject to dropout), and each client independently receives a submodel, so in that perspective, dropout is fundamentally no different, and thus it is a corollary that follows.
>
> The dataset neighboring relationship is indeed add-/remove-one, which was briefly mentioned in Section 2.

---

> > ### Comment · Reviewer_j224 · 2025-04-03
> >
> > Thank you. This, combined with the explanation of the "baseline" in the other rebuttals addresses most of my concerns, particularly w.r.t. experimental evaluation of the bounds and the resultant privacy--utility trade-off.
> >
> > ---
> >
> > I just have one more question w.r.t. dropout:
> > Theorem 3.4 assumes that we have a disjoint partitioning into submodels. Theorem 3.5 assumes that we sample a disjoint partitioning. Given such a partitioning, each client is assumed to independently sample one of the disjoint submodels.
> >
> > However, many of the $2^m$ submodels will be non-disjoint.
> > Could you maybe clarify why Theorem 3.4/3.5 still applies? Can this per-client-dropout somehow be written as a probabilistic mixture of disjoint partitionings?
> >
> > ---
> >
> > Overall, I still think that this is a good submission that extends privacy amplification into an interesting, novel direction (using other sources of randomness than shuffling / i.i.d. sampling of batches for amplification) and continue to recommend acceptance.

---

> > > ### Author Response · Authors · 2025-04-07
> > >
> > > Thank you for following up and clarifying the original concern. The answer is in the proof of Corollary 3.7 in the appendix, but we can give an overview here. Dropout with rate $0.5$ can indeed be seen as a probabilistic mixture of disjoint partitionings. For any subnet $W_J$ formed by dropout, the complementary subnet $W_J^c$ (i.e., the subnet formed by the parameters not in $W_J$) has the same probability to be chosen as $W_J$ because we require dropout with probability $0.5$ (both $W_J$ and $W_J^c$ have probability $2^{-m}$ where $m$ is the number of parameters subject to dropout). This forms two disjoint submodels, so in Corollary 3.7, $d=2$. Dropout is then a probabilistic mixture of these over all potential $W_J$'s. Remark 3.6 also applies here, i.e., there may be a tighter way to leverage the randomness in dropout, but it is still a significant improvement over ignoring randomness entirely and defaulting to the standard DP-SGD analysis.

---

### Official Review · Reviewer_iUZc · 2025-03-10

**Overall Recommendation:** 3

**Summary:**

The paper shows how Renyi DP guarantees can be amplified under two different kinds of data sub-sampling and partitioning strategies. The first is where all data points are used for the same number of iterations but in randomly distributed steps. The second is where different parts of the model are updated with randomly chosen data samples. In each case, the paper shows when the amplification is larger than what was previously known by existing bounds.

**Claims And Evidence:**

Yes, I found most claims to be well-managed in terms of what is claimed and what is shown. The following cases were a bit problematic for me though it was not overclaiming per se -

1. That the amplification with model partitioning  can be helpful for federated and distributed learning- (see Line 074 left column), however there were no experimental setups proposed to show this. Given that the benefit large depends on the exact problem parameters it is hard to justify if this indeed results in a benefit and is not just a theoretical construct.

2. Similarly for Balanced iteration subsampling, the introduction promised that balanced subsample iteration can overcome the issue of disparate sampling in poisson subsampling and can be used in both centralised and federated learning, but there wasn't an experimental validation of this (and the regime where advantage is shown to exist in Fig 2 and 3 were very limited)

**Essential References Not Discussed:**

Well done.

**Experimental Designs Or Analyses:**

This is severely lacking as I have mentioned above.

**Methods And Evaluation Criteria:**

Evaluation is severley lacking. See above for claims that would be good to verify.

**Other Comments Or Suggestions:**

Update: I have updated my score to weak accept.

**Other Strengths And Weaknesses:**

1. The paper is very well written and with one reading it was very clear to me - what the gap in the literature is, how the authors are solving it, what the theoretical statements and roughly how they are proven.

2. The technical novelty is not particularly large but it is still interesting in terms of the results that are proven. The problem setting is also itneresting.

3. There are three major weaknesses.
    i. The theoretical regimens of improvement, at least what is evident from the paper, is very limited. For example, if I have interpreted  Line 270 to 274 correctly (correct me if I am wrong) and if I make a parallel to classical non-private training, $\gamma$ should be thought of as $1/\mathrm{batch size}$ which is the RV indicating what fraction of iterations have that particular example. Now this is of the order of $1/256\ll 0.005$. In this regime Poisson subsampling seems to be better. (I understand that the regime where the binomial heavily concentrates around its mean is where poisson subsampling should be better and this is precisely the case above but also this is the common regime)
    ii) The paper seems to be incomplete in the sense that several things could be improved ( also highlighted by the author). For example, Theorem 3.5 only uses one of the two sources of randomness, Line 196 also seems to be a very loose way of going about it, and Line 305.

    iii) Pending the above and the limited theoretical contribution, I am not viewing its theoretical contribution to be large enough to exempt the need for sufficient experimental evidence to validate its contribution. In this space, the paper's results are very limited.

**Questions For Authors:**

Please address the above comments.

**Relation To Broader Scientific Literature:**

Well done.

**Theoretical Claims:**

I have not checked the proofs but the theorem statements are very well written and the proof sketches are well motivated. So I do not doubt that they are correct (or can be correct in case there are typos or mistakes)

---

> ### Author Rebuttal · Authors · 2025-04-01
>
> We thank the reviewer for their review and their valuable feedback. We will incorporate their comments in our revised paper.
>
> Following the reviewer's suggestions, we have done additional experiments for both centralized and federated settings. In the centralized setting, training ResNet101 using 3 submodels    achieves 79.80\% accuracy with sample-level $(8, 10^{-5})$-DP guarantee for the final model under the standard DP-SGD analysis, and accuracy increases to 82.43\% for the same privacy guarantee using our analysis (this is because our analysis accounts for the amplification gain for model splitting, allowing us to add less noise while still achieving the same DP guarantee); training ResNet101 with 8 submodels  has 76.80\% accuracy under the standard DP-SGD analysis and using our analysis accuracy increases to 80.52\%. In the federated setting, training ResNet101 using 3 submodels achieves 78.47\% accuracy and user-level $(8, 10^{-5})$-DP under the standard analysis, using our analysis accuracy increases to 80.28\% under the same privacy guarantee; 8 submodels with the standard analysis has 76.96\% accuracy, and using our analysis increases the accuracy to 79.11\%.
>
> We would like to clarify the main concern regarding the comparison between Balanced Iteration Subsampling and Poisson Subsampling. Even though we show that  Balanced Iteration Subsampling can have better privacy guarantees in some regimes, as the reviewer points out Balanced Iteration and Poisson Subsampling achieve similar privacy-accuracy trade-offs experimentally. This is because standard training includes a large number of iterations, so both the training dynamic and privacy guarantees of the two become comparable (see experimental results below). The main advantage of Balanced Iteration is that it is more practical and deployment-friendly especially in the federated setting. Poisson Subsampling has implementation-related drawbacks as it gives a variable load to each client, which can overwhelm the resources of the client and undermine fair use policies. The DP community has been interested in studying other (more deployment-friendly) data subsampling techniques whose utility and privacy guarantees are comparable to Poisson Subsampling, for example random shuffling and random check-ins (Balle et al 2020). Our paper fills this gap by showing that this is indeed the case for Balanced Iteration Subsampling. As for the actual experiments, for CIFAR-10 with WideResNet-40-4,  $(8, 10^{-5})$-DP,  2000 iterations, using each sample 655 times for Balanced Iteration Subsampling and with probability $\frac{655}{2000}$ in each iteration for Poisson Subsampling, Balanced Iteration Subsampling injects noise with $\sigma = 10.17$ and achieves validation accuracy of $70.21\%$ (with standard deviation $0.69\%$), while Poisson Subsampling injects noise with $\sigma = 10.20$ and achieves validation accuracy of $70.13\%$ (with standard deviation $0.78\%$). ResNet-101 gives similar results. We will include these experiments in the revision.
>
> On the second concern, Theorem 3.5 and Corollary 3.7 indeed utilize only one of the two sources of randomness for amplification because different training dynamics require different privacy analyses. While our analysis does not fully capture all sources of randomness in the setting of Thm 3.5 (this would require an extension of our current analysis), we hope the reviewer will appreciate that we demonstrate a significant improvement over ignoring randomness entirely and defaulting to the standard DP-SGD analysis. We hope our paper will inspire the DP community to further explore such inherent gains and develop stronger mathematical tools for their accounting.  This remains a promising direction for future work.
>
> Regarding Line 196, which proposes using two clipping norms—one for the parts of the model with model splitting and another for the parts without—it is, in our view, the only viable approach. If a single clipping norm were used, the worst-case $\epsilon$ would occur when the gradient vector concentrates all its power on the part of the model without model splitting. In that scenario, there would be no privacy amplification at all. Therefore, two separate clipping norms appear necessary to the authors.
>
> On the third point, our paper is not merely about introducing a theoretical tool to quantify privacy in specific cases; rather, we view the broader contribution of our paper as highlighting an important but previously overlooked insight —that inherent randomness in the training dynamics, such as model parallelism techniques already used in centralized and federated learning, can be leveraged for privacy amplification, at no additional cost. To our knowledge, this is the first work to identify and quantify such an amplification gain. Beyond the specific analysis presented, we aim to bring this perspective to the DP community and encourage further exploration of how different training dynamics yield privacy gains.

---

### Official Review · Reviewer_czZi · 2025-03-12

**Overall Recommendation:** 3

**Summary:**

The paper explores how inherent randomness in machine learning training can be used for privacy amplification, specifically model partitioning and data subsampling. These methods can potentially enhance the training privacy without adding excessive noise.

**Claims And Evidence:**

It is somewhat unclear whether the study proposes an amplified privacy scheme or a tighter privacy analysis. The current claim leans towards the former, but it lacks a thorough privacy-utility tradeoff analysis and convergence analysis. This leaves open the question of whether the proposed stronger privacy comes at the cost of worse utility or convergence rate compared to the canonical DP-SGD method.

A finer concern is the intuition behind the proposed balanced data subsampling. As it fixes the number of iterations each sample appears in, the proposed balanced sampling introduces less randomness than Poisson sampling. Does this balanced sampling improve only the worst-case privacy, or does it also improve the average privacy across all samples?

**Essential References Not Discussed:**

The related works are properly cited.

**Experimental Designs Or Analyses:**

The current experiments do not address the comparison of the privacy-utility tradeoff between the proposed techniques and the existing DP-SGD training pipeline. Further experiments are needed to address how the privacy amplification methods affect the training performance, especially in convergence and model utility.

In addition, for the balanced sampling scheme, the experiment setup should elaborate on whether it evaluates the average or worst-case privacy of the proposed and baseline sampling methods, and how.

**Methods And Evaluation Criteria:**

The proposed methods including model and data partitioning are relevant and make sense for privacy amplification in ML model training.

**Other Comments Or Suggestions:**

The reviewer suggests adding more experiments with different model structures, training datasets and training hyperparameters.

**Other Strengths And Weaknesses:**

The reviewer suggests clearly stating the baseline privacy analysis for a more direct comparison. It is currently only represented by a single line in the experimental figures, with no rigorous equations or specific experimental setups. It would be helpful to clearly state the baseline privacy guarantee, and mathematically compare it with the proposed Theorems 3.4, 3.5, and 3.8 to directly demonstrate the advantages of the proposed schemes.

The manuscript would benefit from better organization, particularly in making the theoretical analysis more intuitive. The proof of Theorem 3.1 could be summarized as a helper lemma in the main text, with full details in the appendix. Each model splitting and data sampling method should have a clear explanation of how it corresponds to Helper Lemma 3.1. For example, the connection between model splitting and the mixed Gaussian distribution in Theorem 3.1 should be explicitly stated. While Remark 3.3 touches on this, it lacks rigor and comprehensiveness.

**Questions For Authors:**

- Please clarify whether the paper proposes a tighter privacy analysis or a better privacy scheme, and add a comparison to baselines in terms of mode utility and training convergence (see Claims And Evidence and Experimental Designs Or Analyses).

- Please detail the baseline privacy analysis including theoretical formulations and experimental setups (see Other Strengths And Weaknesses).

- Please clarify whether the proposed balanced sampling scheme improves worst-case privacy or average privacy (see Claims And Evidence).

**Relation To Broader Scientific Literature:**

The key contributions are closely related to prior works on DP, particularly in the context of model parallelism and randomized training processes. The authors build upon existing techniques such as model partitioning and Poisson subsampling to propose a novel approach that leverages randomness inherent in the training process for privacy amplification.

**Theoretical Claims:**

I checked the correctness of the proof outline in section 3.4.

---

> ### Author Rebuttal · Authors · 2025-04-01
>
> We thank the reviewer for their review and  valuable feedback.
>
> We would like to clarify their main question about "whether the proposed
> stronger privacy comes at the cost of worse utility or convergence rate compared
> to the canonical DP-SGD method." The main contribution of our paper is to develop a mathematical analysis that is able to quantify the privacy gain that comes from various sources of randomness that share a common structure. Model splitting is already used in the literature both in the centralized and federated settings. We are not proposing a new technique here; we are pointing out that model splitting has a free privacy gain which has remained unnoticed in the prior literature (which requires nontrivial analysis to quantify). Since we are not changing the training, same utility and convergence rates can now be achieved with better privacy guarantees. Conversely, for the same privacy, we can achieve better utility and convergence rates than canonical DP-SGD since we need to inject less noise (see experimental results below).
>
> Balanced Iteration Subsampling is indeed a new subsampling method we propose. Balanced Iteration Subsampling is more deployment-friendly that Poisson subsampling, which has implementation-related drawbacks as it gives a variable load to each client. This can overwhelm the resources of the client and undermine fair use policies. The DP community has been interested in studying other (more deployment-friendly) data subsampling techniques whose utility and privacy guarantees are comparable to Poisson Subsampling, for example random shuffling and random check-ins (Balle et al). Our paper fills this gap by showing that Balanced Iteration Subsampling has comparable performance to Poisson subsampling (theoretically we can prove slightly better privacy guarantees for Balanced Iteration Subsampling, however in experiments we observe that Balanced Iteration Subsampling and Poisson subsampling have similar performance, see below for details). We will revise our paper to make these points more clear.
>
> Regarding the question about whether we use "worst-case privacy, or average privacy across all samples", we would like to point out that we always use the standard definitions of $(\epsilon, \delta)$-DP and $(\alpha, \epsilon)$-RDP in the literature. The definition of $(\epsilon, \delta)$-DP can be intuitively thought as the privacy leak in the worst $\delta$-fraction of samples, so in that sense it makes intuitive sense that Poisson subsampling can have worse $(\epsilon, \delta)$-DP than Balanced Iteration Subsampling.
>
> The baseline for our model splitting results is the canonical DP-SGD analysis as pointed out by the reviewer. For one iteration, the canonical DP-SGD analysis  would give $\epsilon = \frac{\alpha c^2}{2\sigma^2}$ RDP in contrast to the RDP bound in Theorem 3.1 (this expression for $\epsilon$ does not have a dependence on the number of submodels because the canonical analysis does not leverage the privacy amplification gain due to model splitting, which is the contribution of our paper). Figure 1 presents a visual comparison of our analysis and the standard analysis. We will make this point clear in the revision. To put this into perspective, for $(\epsilon, \delta)$-DP with $\delta = 10^{-5}$, 1200 training iterations, data subsampling rate of 0.1, a noise standard deviation of 2, our analysis can guarantee $\epsilon = [3.5, 4.0, 5.0, 7.4]$ for $[8, 6, 4, 2]$ submodels, while without using our analysis, the best guarantee is $\epsilon = 11.0$ for all numbers of submodels. To give another example, with $\sigma = 6$, our analysis guarantees $\epsilon = [1.0, 1.2, 1.5, 2.1]$ while standard analysis gives $\epsilon = 3.0$. We will add these as graphs to the revised version. Note that because training remains the same and we only change the privacy analysis, utility remains the same in both cases.
>
> Conversely, we can fix the same $(\epsilon, \delta)$-DP budget and compare utility. We have done such experiments for both centralized and federated settings. In the centralized setting with sample-level $(8, 10^{-5})$-DP,  training ResNet101 with 3 submodels under the standard DP-SGD analysis achieves 79.80\% accuracy and using our analysis accuracy increases to 82.43\% (this is because our analysis accounts for the amplification gain for model splitting, allowing us to add less noise while still achieving the same DP guarantee); 8 submodels under the standard DP-SGD analysis has 76.80\% accuracy and using our analysis accuracy increases to 80.52\%. In federated setting with user-level $(8, 10^{-5})$-DP,  3 submodels with the standard analysis has 78.47\% accuracy and using our analysis accuracy increases to 80.28\%; 8 submodels with the standard analysis has 76.96\% accuracy has using our analysis increases it to 79.11\%.
>
> We will add these experiments to the revised version of the paper, and incorporate reviewer's suggestion about the restructuring of the results.

---

> > ### Comment · Reviewer_czZi · 2025-04-05
> >
> > The clarification that Balanced Iteration Subsampling is primarily deployment-friendly rather than offering a stronger privacy guarantee addresses my main concern. This distinction from the model splitting part, which provides a nontrivial privacy gain, should be clearly stated in the revision. With the above clarification and the additional experiments, I would consider a weak accept.

---

> > > ### Author Response · Authors · 2025-04-07
> > >
> > > Thank you for reading our rebuttal and updating your review!

---

### Official Review · Reviewer_rbZT · 2025-03-17

**Overall Recommendation:** 3

**Summary:**

The paper proposes a unified privacy analysis for the applications of model and data partitions. The crucial theorem as shown in Theorem 3.1, which is novel and non-trivial to my best knowledge, states the renyi-divergence between a Gaussian distribution and a mixture of Gaussians. Upon to this theorem, the privacy analysis for applications such as model splitting, dropout, and balanced iteration subsampling,  can be amplified accordingly. The paper empirically compares the proposed privacy analysis and the analysis in the literature in their experiment.

**Claims And Evidence:**

The claims and evidence are mostly convincing to me. The only place that can be more supported is a straightforward comparison between the proposed analysis and the analysis in the literature. In the application of balanced iteration subsampling, the paper states the privacy guarantee of the analysis in the literature. However, a similar statement for the analysis in the literature is not shown for applications in Section 3.2. Moreover, the numerical comparison is only conducted with one choice of $\sigma$.

**Essential References Not Discussed:**

[1] seems to also study the mixture of Gaussian for Renyi-DP, while it specifically calculates the single-dimensional situation.

[1] Mironov, Ilya, Kunal Talwar, and Li Zhang. "R\'enyi differential privacy of the sampled gaussian mechanism." arXiv preprint arXiv:1908.10530 (2019).

**Experimental Designs Or Analyses:**

1. The empirical evaluation is conducted only with one dataset and model. As the comparison between the analysis in the literature and the proposed literature is not consistent for all cases (of hyperparameter configurations), empirical evaluation is necessary to show the effectiveness of the proposed analysis.
2. Only one application, model partitioning, is evaluated in the experiment section. Other applications, such as dropout and balanced iteration subsampling, are not evaluated.

**Methods And Evaluation Criteria:**

The algorithms and the theoretical results make sense.

**Other Comments Or Suggestions:**

N/A

**Other Strengths And Weaknesses:**

N/A

**Questions For Authors:**

The motivation statements (line 158-164) in Section 3.2.2 are not clear. For the blocks that are sensitive to pruning, why is it better to not partition them?

**Relation To Broader Scientific Literature:**

N/A

**Theoretical Claims:**

I have checked the proof of the theorems for the applications (Theorem 3.4 - 3.8), but have not checked the exact proof of Theorem 3.1 (in appendix).

---

> ### Author Rebuttal · Authors · 2025-04-01
>
> We thank the reviewer for their review and their valuable feedback.
>
> We would like to clarify their main concern regarding "a straightforward comparison between the proposed analysis and the analysis in the literature" for the  model splitting methods in Section 3.2. We note that our paper is the first one to point out that model splitting has an inherent privacy amplification gain and to quantify this privacy gain. The baseline to compare this is the standard RDP analysis  for DP-SGD which does not take this amplification gain into account since it has remained unnoticed and unquantified in the literature. For one iteration, the standard analysis in the literature would give $\epsilon = \frac{\alpha c^2}{2\sigma^2}$ in contrast to the RDP bound in Theorem 3.1 (this expression for $\epsilon$ does not have a dependence on the number of submodels, because again it does not take into account the fact that training uses model splitting).  Figure 1 presents a visual comparison of our analysis and the aforementioned standard analysis. We will make this point clear in the revision. To put this into perspective, for $(\epsilon, \delta)$-DP with $\delta = 10^{-5}$, 1200 training iterations, data subsampling rate of 0.1, a noise standard deviation of 2, our analysis can guarantee $\epsilon = [3.5, 4.0, 5.0, 7.4]$ for $[8, 6, 4, 2]$ submodels, while without using our analysis, the best guarantee is $\epsilon = 11.0$ for all numbers of submodels. To give another example, with $\sigma = 6$, our analysis guarantees $\epsilon = [1.0, 1.2, 1.5, 2.1]$ while standard analysis gives $\epsilon = 3.0$. We will add these as graphs to the revised version.
>
> Conversely, we can fix the same $(\epsilon, \delta)$-DP for the final model and compare accuracy. We have done such experiments for both centralized and federated settings. In the centralized setting with sample-level $(8, 10^{-5})$-DP,  training ResNet101 with 3 submodels under the standard DP-SGD analysis achieves 79.80\% accuracy and using our analysis accuracy increases to 82.43\% (this is because our analysis accounts for the amplification gain for model splitting, allowing us to add less noise while still achieving the same DP guarantee); 8 submodels under the standard DP-SGD analysis has 76.80\% accuracy and using our analysis accuracy increases to 80.52\%. In federated setting with user-level $(8, 10^{-5})$-DP,  3 submodels with the standard analysis has 78.47\% accuracy and using our analysis accuracy increases to 80.28\%; 8 submodels with the standard analysis has 76.96\% accuracy has using our analysis increases it to 79.11\%.
>
> As for the comparison between Balanced Iteration Subsampling and Poisson Subsampling, we would like to emphasize a point which we will make more clear in the revision. Balanced Iteration and Poisson Subsampling achieve similar privacy-accuracy trade-offs experimentally. This is because standard training includes a large number of iterations, and for large numbers of iterations, both the training dynamic and privacy guarantees of the two become comparable as noted in the paper. The main advantage of Balanced Iteration Subsampling is that it is more practical and deployment-friendly especially in the federated setting. Poisson Subsampling has implementation-related drawbacks as it gives a variable load to each client, which can overwhelm the resources of the client and undermine fair use policies. The DP community has been interested in studying other (more deployment-friendly) data subsampling techniques whose utility and privacy guarantees are comparable to Poisson Subsampling, for example random shuffling and random check-ins (Balle et al 2020). Our paper fills this gap by showing that this is indeed the case for Balanced Iteration Subsampling. As for the actual experiments, for CIFAR-10 with WideResNet-40-4,  $(8, 10^{-5})$-DP,  2000 iterations, using each sample 655 times for Balanced Iteration Subsampling and with probability $\frac{655}{2000}$ in each iteration for Poisson Subsampling, Balanced Iteration Subsampling injects noise with $\sigma = 10.17$ and achieves validation accuracy of $70.21\%$ (with standard deviation $0.69\%$), while Poisson Subsampling injects noise with $\sigma = 10.20$ and achieves validation accuracy of $70.13\%$ (with standard deviation $0.78\%$). ResNet-101 gives similar results. We will include these experiments in the revision.
>
> While we are aware of the paper by (Mironov et. al.) which also considers Poisson subsampling with the Gaussian mechanism; we use the stronger analysis in (Zhu \& Wang) for Poisson subsampling.
>
> Lastly, the design choice to not partition the first and last layers of ResNet-101 is guided by insights from the model splitting literature (e.g. Dun et. al.). They argue that since these layers pick up important features, partitioning them leads to degraded accuracy. Following the guidance from this literature, we experimented with partial and not fully disjoint model splitting.

---

### Decision · Program_Chairs · 2025-05-01

**Decision:**

Accept (poster)

**Comment:**

The paper proposes Renyi DP accounting for new DP-SGD variants: privacy amplification from updating random partitions of the model and accounting for balanced iteration subsampling.

Both of these as well as the underlying bound on Renyi divergence between a Gaussian and a mixture-of-Gaussians are interesting and valuable contributions.

All reviewers recommend acceptance, with main criticism related to limited experiments. The authors have added new experiments in the rebuttal, which seems sufficient for a theory-focused paper.

A minor weakness noted by Reviewer j224 and not addressed by the authors is the lack of comparison with state-of-the-art numerical accountants for Poisson subsampling (such as the PRV accountant that is the standard e.g. in Opacus). This would be necessary to fully judge the value of the proposed methods compared to state-of-the-art in $(\varepsilon, \delta)$-DP.